# From surviving to thriving: How sleep, physical activity, and diet shape well-being in young adults

Jack R.H. Cooper[1], Robin S. Turner[2], Tamlin S. Conner [3]*

**1** Department of Psychology, University of Otago, Dunedin, New Zealand, **2** Department of Preventative and Social Medicine, University of Otago, Dunedin, New Zealand, **3** Department of Psychology, University of Otago, Dunedin, New Zealand

* tamlin.conner@otago.ac.nz

## Abstract

Healthy lifestyles are a cornerstone of optimal physical health, yet their contribution to optimal mental health is still an open question. This article investigated the relationships between three key health behaviors—sleep quality, physical activity, and dietary choices—and psychological well-being among young adults ages 17–25, a demographic known for disproportionately low levels of well-being. The research used three datasets: a cross-sectional survey (Study 1, N = 1,032) and two daily diary datasets (Study 2, N = 818; Study 3, N = 236) of young adults living in New Zealand, the United Kingdom, and the United States. Multilevel regression analyses examined how these health behaviors simultaneously and synergistically predicted well-being at the between-person and within-person levels, while controlling for covariates such as age, gender, socioeconomic status, and depressive symptoms. Additionally, Study 3 incorporated wearable device-measured physical activity to supplement self-report measures. The findings highlighted better sleep quality as the strongest predictor of well-being across all three datasets, followed by fruit and vegetable consumption, with significant associations between these health behaviors and well-being observed at the between-person and within-person levels. Physical activity also emerged as a reliable predictor of well-being, mostly at the within-person level. The use of device-measured physical activity confirmed similar patterns, reinforcing the validity of findings. Pathways linking health behaviors to well-being were mostly additive, except for a buffering pattern observed at the within-person level indicating that higher fruit and vegetable intake could potentially mitigate the negative impact of poor sleep on daily well-being. This article underscores the importance of the "big three" health behaviors in well-being among young adults, offering insights for future health interventions to improve positive psychological functioning in this population.

**Data availability statement:** All files are available from the Open Science Framework: https://osf.io/gs8y7/.

**Funding:** Portions of this research were funded by the New Zealand Health Research Council Emerging Researcher First Grant to TS Conner (Grant #12/709); https://www.hrc.govt.nz/ Funders played no role in the study design, data collection or analysis, decision to publish, or preparation of this manuscript.

**Competing interests:** The authors have declared that no competing interests exist.

## Introduction

Over the last century, psychological well-being—encompassing positive emotions, satisfaction with life and positive psychological and social functioning—has emerged as an essential element of mental health, reflecting positive states of health distinct from negative mental health disorders [1–4]. Psychological well-being has been linked to a variety of positive outcomes including resistance to stress [5–8], improved physical health and longevity [9–16] and creativity in the workplace [17–21]. Young adults (ages 17–25; [22]) are disproportionately likely to exhibit low levels of well-being due to a variety of factors including, but not limited to, poor economic support, modern pressures including social media, climate change concerns, disrupted support systems from leaving the parental home, pressures to establish autonomy, and unhealthy lifestyles common to this age group [22–24]. This unique vulnerability means that research focusing on how best to facilitate well-being in young adults is vital.

Given increasing appreciation of the connection between mental and physical health [25], one growing area of research focuses on a relatively understudied collection of factors that contribute to well-being – the role of health behaviors. Health behaviors refer to modifiable actions that individuals can implement in their daily lives that are associated with better health outcomes. Three health behaviors have emerged as particularly promising predictors of well-being, being dubbed the "big three" [26]. They are high-quality sleep [27–29,30–33], physical activity [34–38], and a healthy diet [39–41]. The associations between the big three health behaviors and well-being are both biologically and psychologically plausible. For example, poor quality sleep has been linked to immediate disruptions in the human body's circadian rhythm, which is known to negatively impact mood including a reduction in positive mood [42,43]. Similarly, physical activity has been linked to increased serotonin levels (itself linked to improved mood; [44,45]) and to improved feelings of control (a core component of positive functioning; [46]). And, a healthy diet rich in fruits and vegetables provides important nutrients such as vitamin C/B and complex carbohydrates, which have been linked to positive mood, feelings of vitality, and creativity [47–49]. In contrast, high consumption of ultra-processed foods has been linked to negative outcomes for both phyiscal and mental health [50,51,52–55].

However, existing research on the role of health behaviors for young adult well-being is limited in three key ways. First, health behavior research often focuses on links with mental illness, namely whether health behaviors hold potential for reducing risk and alleviating mental symptoms [56]. However, recent models of mental health suggest that it is more than just the presence or absence of mental illness. The dual-continuum model is one such model that posits that well-being and mental illness are distinct yet related dimensions, rather than opposite ends of a single spectrum. This model suggests that individuals can experience low mental illness without necessarily having high well-being, and vice versa. Empirical support for this framework has grown, with studies demonstrating its superiority over single-factor models [1–3,57]. For instance, a scoping review by Iasiello et al. (2020) found that 82 out of 83 studies provided evidence supporting the dual-continuum perspective. Given its conceptual distinctness, research investigating how health behaviors associate with well-being over and above mental illness is crucial.

Second, research typically tests health behaviors in isolation of each other without considering their differential, additive, or possibly synergistic contributions to well-being. Research testing multiple health behaviors is rare, and the research that does exist each have their own limitations from cross-sectional data [26,58] to only testing two health behaviors [27,59,60] to small sample sizes (n = 84; [61]). As such, research examining all three health behaviors in large datasets is required to gain a better understanding of their unique associations with well-being, as well as any potential interactions between them (e.g., a synergistic interaction where good sleep leads to greater benefits from physical activity). To date, there is some evidence to support two-way interactions between the health behaviors in predicting mental health only [26,58,59]. For example, Wickham et al. (2020) surveyed the big three health behaviors together in a cross-sectional survey and found some evidence of a protective interaction where fruit and vegetable consumption mitigated some of the detriments associated with poor sleep and/or physical activity, while Isanejad et al. (2016) found evidence for a synergistic interaction where those with a higher Baltic Sea Diet score (indicating a greater intake of whole grains, vegetables, fruits and fish such as salmon and herring) saw greater benefits to their life satisfaction scores from engaging in ≥ 2.5 hours of physical activity a week. However, statistical interactions or patterns of moderation are often not reliable or reproducible, or are underpowered [62]. This means they require replication in other datasets to confirm their stability and usefulness for potential intervention targets.

Third, past research has often used cross-sectional data that only focuses on between-person differences. Between-person differences answer research questions about different individuals, such as "does John, who sleeps better than Jane, report higher well-being?". However, questions regarding within-person differences, such as "does John report higher well-being on days where he slept well compared to days where he slept poorly?" are often of great interest to psychology [63], as they can help answer the implicit question of "will the same person benefit from changing what they are doing?" Micro-longitudinal or intensive longitudinal data [64] is ideally suited to investigating within-person patterns. This type of data involves repeated observations from the same individual over time, often derived through daily diary designs or methods like ambulatory assessment, ecological momentary assessment, and experience sampling methods [65,66]. While there has been daily diary research examining within-person associations between health behaviours and well-being [67–75], it is rare that analyses acknowledge and partition both between-person *and* within-person associations to be discussed separately. As such, research using repeated measures datasets and statistical techniques that can distinguish between these two levels (between-person and within-person) has the potential to inform a more complete understanding how these health behaviors are associated with well-being.

The goal of the present paper was to investigate how the big three health behaviors of sleep quality, physical activity, and diet predicted psychological well-being among young adults using both cross-sectional and daily diary methods. The current paper aimed to advance research in three key ways by: (i) examining well-being as a primary outcome, controlling for depressive symptoms, to isolate the associations between health behaviors and positive psychological functioning; (ii) investigating the health behaviors simultaneously as predictors of well-being, and consider their potential additive or interactive associations in predicting well-being, and; (iii) incorporating daily diary techniques to measure health behaviors and well-being repeatedly over time to separate the health behavior-well-being associations into their between-person and within-person components.

## Methods

### Overview of datasets

The three datasets included one cross-sectional dataset and two daily diary datasets together surveying more than 2,000 young adults ages 17–25 living in New Zealand, the United States and the United Kingdom (UK), all English-speaking Western-Educated-Industrialized-Rich-Democratic (WEIRD) countries [76]. Study 1 was a cross-sectional study of 1,032 young adults living in New Zealand, the United States, and the United Kingdom called the Lifestyle of Young Adults Survey. Study 2 was a 13-day daily-diary study of 818 young adults living in New Zealand called the Daily Life Study, which has been used previously to investigate the health of young adults [19,77,78], though these studies either examined other

predictors than health behaviors (e.g., creative activity; [19,77]) or only examined between-person associations without regard for within-person associations [78]. Study 3 was an 8-day daily diary study of 236 young adults living in New Zealand, which also included device-measured tracking of physical activity using Fitbits. Each dataset contained survey data that included demographic characteristics, an extensive battery of health behaviors including sleep quality, physical activity, and dietary habits, and measures of mental health and well-being. Analyses were similar across the three studies and investigated the between-person (Studies 1–3) and within-person (Studies 2 and 3 only) associations between the health behaviors and well-being, adjusting for covariates.

## Participants and procedures

The three datasets analyzed in this study had been previously collected, and were accessed for research purposes from April 26th, 2024 onwards. The authors did not have access to information that could identify individual participants during or after data collection. All three studies used convenience sampling to collect participants, but Study 1 stratified recruitment by gender to achieve a more equal gender balance between men and women.

**Study 1.** The Lifestyles of Young Adults Survey was a cross-sectional survey conducted over multiple years (2017, 2018, 2019, and 2021) approved by the University of Otago Department of Psychology (Ethics #D17/158), with oversight by the University of Otago Ethics Committee. We used data from the 2021 survey because prior publications used data from earlier waves [26]. The survey was restricted to young adults ages 18–25, and collected information on their demographic characteristics, health behaviors, and psychological well-being (See Measures, below). In 2021, participants were recruited from New Zealand via the University of Otago, the United States via Amazon's Mechanical Turk (MTurk) and the United Kingdom (UK) via Prolific. Participants had to meet the following criteria: (i) be 18–25 years old at the time of participation; (ii) pass two attention checks (for those completing the survey via MTurk or Prolific); (iii) reside in the United States (for MTurk) or the UK (for Prolific) and; (iv) not have completed the survey in previous years (automatic exclusion).

Participants accessed a study website to provide informed consent and complete the 30-minute online survey. New Zealand participants were recruited through the Department of Psychology at the University of Otago and awarded with course credit. Male students were additionally recruited via flyers and compensated with $10 to achieve gender balance. American MTurk participants were compensated an average of US$3.25, while British Prolific participants received £3.50. Data collection took place between May 2021 and November 2021.

For analysis, 1,032 young adults (450 men; 43.6%) aged 18–25 years old (M = 22.07) from New Zealand (n = 346, 33.6%), the United States (n = 343; 33.2%), and the United Kingdom (n = 343; 33.2%) provided usable data. Additional participants were excluded from this analysis for not passing attention checks (n = 95), being outside the age range (n = 19), discontinuing the survey (n = 12), and showing evidence of response bias (n = 8).

**Study 2.** Study 2 used archival data from the 2013 and 2014 waves of the Daily Life Study, a daily diary study conducted over multiple years (2011, 2012, 2013, and 2014) approved by the University of Otago Ethics Committee (Ethics #10/177). Data from the 2011 and 2012 waves were not used as it did not contain dietary data. The Daily Life Study included an initial survey with an extensive battery of demographic characteristics, followed by a 13-day daily diary survey of a range of their daily health behaviors and well-being (see Measures, below). Participants had to meet the following criteria: (i) be 17–25 years old at the time of participation, and (ii) own a mobile phone.

Participants attended an initial session in person within the Department of Psychology at the University of Otago. During this session, participants signed informed consent and completed a background survey on demographic and trait characteristics (e.g., age, gender, ethnicity, personality) over the computer. After receiving instructions for the daily diary portion of the study, participants began completing an online daily diary survey accessible only between 3 pm and 8 pm starting the next day. This survey covered topics such as health behavior engagement and daily well-being. After 13 days of completing daily diary surveys, participants returned to the lab for debriefing and compensation.

For analysis, 818 young adults (227 men; 27.8%) aged between 17 and 25 years old (M = 19.73) provided usable data. Additional participants were excluded for having 0 daily diary records (N = 9).

**Study 3.** Study 3 used data from the Fitbit Study, a daily diary and activity-tracking study conducted from 2021–2022 approved by the University of Otago Department of Psychology (Ethics #D21/156), with oversight by the University of Otago Ethics Committee. Similar to Study 2, this was an observational daily diary design that included an initial survey with demographic characteristics, followed by an 8-day daily diary survey of a range of daily health behaviors and well-being, alongside device-measure activity tracking using a Fitbit.

Participants attended an initial session in person within the Department of Psychology at the University of Otago. During this session, participants gave informed consent, completed a background survey regarding demographic and trait characteristics on the computer, and provided their mobile phone numbers to receive survey links each day. Due to a COVID-19 outbreak in New Zealand in late 2021 and the resulting restrictions on in-person research, some participants completed the majority of this initial session remotely online. Participants had to meet the following criteria: (i) be aged 18–25; (ii) currently enrolled in either a 100-level or 200-level psychology course at the University of Otago and; (iii) have a working smartphone. After receiving instructions on the daily diary study, participants began completing two surveys a day, namely a morning survey link sent via text at 10am each day and an evening survey link sent via text at 10 pm each day. Morning and evening surveys were programmed in Qualtrics. The morning survey assessed self-reported sleep quality, among other measures not relevant to this report, while the evening survey assessed the other health behaviors enacted that day (i.e., physical activity and dietary behaviors) and well-being for that day (see Measures below). Participants were additionally lent a Fitbit Inspire 2 [79] at the initial session, which they were instructed to wear on their wrist at all times during the study to monitor their physical activity. After 7 days, the participants returned to the lab to be debriefed, return their Fitbit and complete an exit survey. For analysis, 227 young adults (54 men; 23.8%) aged 18–25 (M = 19.37) provided usable data. Additional participants were excluded for failing to attend the initial session (N = 9).

## Measures

**Covariates.** All three datasets included similar measures of covariates related to: (i) age; (ii) gender; (iii) ethnicity; (iv) childhood socioeconomic status (SES); and (v) body mass index (BMI). Across all three studies, age was assessed via a single item asking participants to select their age ranging from 18 to 25+ (Study 1 & 3) or 17–25+ (Study 2). Gender was assessed either as a binary choice of male or female (Study 2) or a three-category choice of male, female, or gender diverse (Studies 1 & 3). Ethnicity was assessed as a multiple-option measure where participants could select what ethnicities applied to them. Childhood SES was assessed using a 3-item measure in all three studies [80] (αs = 0.787 to 0.824) that asked participants to indicate their agreement on a 7 point scale (1 = strongly disagree, 7 = strongly agree). The 3 items were: (a) "My family usually had enough money for things when I was growing up"; (b) "I grew up in a relatively wealthy neighborhood"; (c) "I felt relatively wealthy compared to the other kids in my school." Body Mass Index (BMI) was assessed either by self-reporting height and weight (Studies 1 & 3) or by measuring height and weight (Study 2) and computing BMI by the standard equation of weight (kg)/height$^2$ (cm).

In addition, all three datasets included a further covariate measuring (vi) depressive symptoms at both the between-person level and the within-person level where relevant (i.e., Study 2 and 3). Depressive symptoms was included as a covariate given it is the best proxy measure of mental ill-health [81], and as such, the most important to be held constant. Depressive symptoms were assessed at the between-person level using the 8-item PROMIS Emotional Distress Depression and Anxiety Short Form 8a scales [82] (Studies 1 & 3) or the 20-item Center for Epidemiological Depression Scale (CES-D) [83] (Study 2), all with excellent reliability (αs = 0.892 to 0.937). At the within-person level, depressive symptoms was assessed using a proxy measure of daily negative mood: For Study 2, a 3-item low activation negative mood scale asked participants to report how "dejected/sad/unhappy" they felt today, on a scale from 1 (Not at all) to 5 (Extremely)

averaged each day (within-person Omega reliability or $\Omega = 0.780$, calculated using the user written command "omegaSEM" in R; [84]); and, for Study 3, a 3-item Depression subscale from the Profile of Mood States (POMS) [85] asked participants to report "Today I felt… sad/discouraged/hopeless" on a scale from 0 (Not at all) to 4 (Extremely) averaged each day ($\Omega = 0.711$).

**Health behaviors.** The "big three" health behaviors related to sleep, physical activity, and diet were chosen for this study because of their importance for health, as determined by the World Health Organization (WHO; [76]). Although the WHO also identified smoking and alcohol intake as important to health, behaviors related to addiction or substance were outside of the scope of this paper. Addictive behaviors have been examined extensively by previous research, and their inclusion here would lead to a significant increase in model terms and risk potential overfitting. For sleep, we focused on *sleep quality*, as evidence suggests it is the stronger predictor of well-being than sleep quantity [26,86–88]. Sleep quantity was also included as a covariate for completeness. For physical activity, we focused on moderate-vigorous activity due to its importance for health as stated by both the Centre for Disease Control (CDC; [89]) and the WHO [90]. We also supplemented the self-report measures using Fitbit-measured physical activity that captured minutes spent performing moderate-vigorous physical activity in Study 3. For dietary behavior, we chose two measures: fruit and vegetable (FV) consumption and ultra-processed foods (UPF) consumption, as both are considered core elements of a healthy diet (i.e., high FV consumption, low UPF consumption; [91,92]).

For sleep quality, Study 1 used the 8-item Patient Reported Outcomes Measurement Information System (PROMIS) Sleep Disturbance Short Form [93], which has good psychometric properties in adult populations [94]. The items assessed a variety of concepts relating to sleep quality, such as "In the past 7 days... My sleep was refreshing: (5) Not at all; (4) A little bit; (3) Somewhat; (2) Quite a bit; and (1) Very much; and "My sleep was restless: (1) Not at all; (2) A little bit; (3) Somewhat; (4) Quite a bit; and (5) Very much". Responses to these items were summed ($\alpha = 0.865$) and converted into T scores via the process recommended by the developers of the measure, resulting in a T-score distribution with a mean of 50 and standard deviation of 10. Higher scores indicated greater sleep disturbance (poorer sleep quality). Sleep quantity was assessed using a single self-report item asking participants "During the past 7 days, how many hours of actual sleep did you get at night? (This may be different than the number of hours you spend in bed)." Participants used a drop-down menu to select the hours per night in 30-minute increments.

Study 2 measured sleep quality and quantity in the daily diary using single face valid items. Sleep quality was assessed by asking participants about their feeling of waking up refreshed, using the question: "Please rate the extent to which you experienced any of the following physical states today- I felt refreshed when I woke up this morning: (0) Not at all; (1) A little; (2) Somewhat; (3) Moderately; (4) Very." Sleep quantity was assessed by asking participants "Approximately how many hours sleep did you get last night?" followed by a drop-down menu to select the hours per night in 30-minute increments up to 15 + hours as the maximal option.

Study 3 measured sleep quality in the morning diary using the 4-item variation of the PROMIS Sleep Disturbance Short Form [93; Sleep Disturbance – Short Form 4a version 1], adapted to a daily format by changing the time frame from "In the past 7 days" to "last night". In each morning survey, participants reported on the quality of their sleep, responding to items such as "Last night, my sleep was refreshing: (5) Not at all; (4) A little bit; (3) Somewhat; (2) Quite a bit; and (1) Very much. As with Study 1, responses to these items were summed ($\Omega = 0.795$) and converted into T scores via the process recommended by the developers of the measure, resulting in a T-score distribution with a mean of 50 and standard deviation of 10. Higher scores indicated greater sleep disturbance (poorer sleep quality). Sleep quantity was measured by a single item asking participants "How many actual hours of sleep did you get last night?" with options ranging from 0 hours to 24 hours in 30-minute increments.

Prior to analyses, the two PROMIS measures of sleep disturbance(Study 1 and 3) were reversed so that higher scores indicated better sleep quality. This was done to allow easier comparison and visualizations across the three studies.

For physical activity, Study 1 used the 7-item International Physical Activity Questionnaire Short Form (IPAQ-SF), which has shown good psychometric properties in adult populations [95]. The items asked participants to report both how many days in the last week they had engaged in walking, moderate physical activity, and vigorous physical activity, such as "During the last 7 days, on how many days did you do vigorous physical activities like heavy lifting, digging, aerobics, or fast bicycling?" or "During the last 7 days, on how many days did you walk for at least 10 minutes at a time?", and how much time they spent doing each category on these days. Average daily physical activity was calculated by multiplying the number of days participants performed moderate and vigorous physical activity by how long they spent doing it on those days, before dividing the number by 7 (e.g., 4 days x 40 minutes on those days/ 7 days = 22 minutes of physical activity a day).

Study 2 measured daily physical activity in each daily diary using a single item based on the IPAQ short-form format [94]: "How much time did you spend doing vigorous and moderate physical activities TODAY? _________minutes (please write a number between 0 and 999)". Prior to answering, participants read instructions to "think about the activities you did today at work, as part of your house and yard work, to get from place to place, and in your spare time for recreation, exercise or sport. Think about all the vigorous and moderate activities that you did. Vigorous physical activities refer to activities that take hard physical effort and make you breathe much harder than normal. Moderate activities refer to activities that take moderate physical effort and make you breathe somewhat harder than normal. Think only about those physical activities that you did for at least 10 minutes at a time."

Study 3 measured physical activity in each evening diary using an adapted 4-item version of the IPAQ-SF to assess time spent sitting, walking, being moderately physically active, and being vigorously physically active each day. Response options ranged from "0 minutes" to "60 minutes" in 15-minute increments, and then in 30-minute increments after that. Responses to the moderate activity and the vigorous activity items were summed together to represent time spent being either moderately or vigorously active. Additionally, the Fitbit devices lent to participants in this study provided objective data regarding their moderate and vigorous physical activity in minutes for each day. The algorithms used to determine physical activity by the Fitbit are proprietary, but "it has been assumed that these categories approximately correspond to light (1.6-2.9 METs), moderate (3.0-5.9 METs), and vigorous (≥ 6.0 METs) physical activity" [96, p4].

For fruit and vegetable (FV) consumption, Study 1 used four modified items from the New Zealand Nutrition Survey [97]. Consumption of raw fruit, cooked fruit, raw vegetables, and cooked vegetables were each assessed via survey items such as "How many servings of raw fruit – fresh or frozen – do you eat per day? For example banana, apple, orange, kiwifruit, berries. Please include any frozen fruit if eaten raw (such as in smoothies). Do not include fruit juice or dried fruit. A serving is the same as a medium piece of fruit such as 1 apple, two small pieces of fruit such as two apricots, or ½ cup of berries (fresh or frozen)". Response options ranged from 0 to 4+ in 0.5 serving increments, with examples of what constituted a serving provided within the question. Daily FV consumption was calculated by summing responses to the four items (raw fruit, cooked fruit, raw vegetables, cooked vegetables).

Study 2 measured FV consumption in each daily diary using two modified items from the New Zealand Nutrition Survey asking participants how many standard servings of fruits and vegetables they ate last night and today (e.g., "How many servings of fruit (fresh, frozen, canned or stewed) did you eat TODAY? Do not include fruit juice or dried fruit. 1 'serving' = 1 medium piece of fruit (e.g., apple) or 2 small pieces of fruit or ½ cup stewed fruit, e.g., 1 apple + 2 small apricots = 2 servings", "How many servings of vegetables (fresh, frozen, or canned) did you eat TODAY? Do not include vegetable juices or hot chips. 1 'serving' = 1 medium potato/kumara or ½ cup cooked vegetables or 1 cup of salad. e.g., 2 medium potatoes + ½ cup peas = 3 servings"). Daily FV consumption was calculated by summing the responses to the fruit and vegetable items.

Study 3 measured FV consumption in each evening diary using the same 4 items as Study 1 (raw and cooked fruits and vegetables), but modified to reflect a daily timeframe in their wording (e.g., "How many servings of cooked, frozen,

or canned/tinned vegetables do you eat TODAY? For example, vegetables cooked in a curry or stew; roast, boiled or steamed vegetables; canned/tinned tomatoes, green beans; frozen veggie mixes. (Do not include hot chips/French fries, kumara chips or deep-fried potatoes; Do not include legumes such as baked beans, kidney beans, chickpeas etc. Do not include raw vegetables)"). Daily FV consumption was calculated by summing the responses to these four items (raw fruit, cooked fruit, raw vegetables, cooked vegetables).

For ultra-processed food (UPF) consumption, Study 1 used seven items created by combining five items from the dietary habit questionnaire in the 2008/09 New Zealand Nutrition Survey [97] and two items from the Short Food Survey from Hendrie et al. (2017; [98]). The items asked participants how many times per week they consumed items from seven different food categories; soft drinks, fast food, French fries, sweets, cakes, potato chips and processed meat (for example, "How often do you eat fast food or takeaways from places like McDonalds etc.? Think about breakfast, lunch, dinner and snacks. Do not include times when you have only purchased a drink/beverage"). Daily UPF consumption was calculated by summing responses to these seven items before dividing the total by 7.

Study 2 assessed UPF consumption in the daily diary using three items asking participants how many standard servings of chips, confectionary items, and soft drinks they had consumed today, with an example of what constituted a standard serving being provided (e.g., "How many servings of lollies, sweets, chocolate, or other confectionary items did you eat TODAY? 1 'serving' = one regular sized chocolate bar (~50g) or 6 jet planes/chocolate eclairs (toffee), the amount that would fit into the palm of your hand"). Daily UPF consumption was calculated by summing the responses to these three items.

Study 3 assessed UPF consumption in the evening diary using two items asking participants how many standard servings of chips or sweets they had consumed today, with an example of what constituted a standard serving (e.g., "How many servings of hot chips, French fries, wedges, or kumara chips did you eat TODAY? Think about lunch and dinner as well as snacks. 1 'serving' = one cup or 1 small/regular fast food serving or ½ scoop of takeaway hot chips. e.g., one scoop takeaways hot chips = 2 servings"). Daily UPF consumption was calculated by summing the responses to these two items.

**Well-being.** Study 1 measured trait well-being using the Flourishing Scale developed by Diener et al. [99]. The Flourishing Scale contains 8 items that ask participants to rate their agreement on a 7-point Likert scale, such as "I lead a purposeful and meaningful life: (1) Strongly disagree; (2) Disagree; (3) Slightly disagree; (4) Neither agree not disagree; (5) Slightly agree; (6) Agree, or; (7) Strongly agree". A raw score was created ranging from 7 (minimum) to 56 (maximum) by summing the scores on these eight items ($\alpha = 0.899$). This raw score was then transformed into a percentage score where 0 indicated the lowest possible summed score for the raw items (i.e., a score of 7) while 100 indicated the highest possible score (i.e., a score of 56). This was done to make comparisons across the well-being measures easier.

Study 2 measured daily well-being in the daily diary using the Flourishing Scale [99] adapted to a daily format. Instructions were modified so that participants reported on their experiences that day (e.g., "Today, I led a purposeful and meaningful life: (1) Strongly disagree; (2) Disagree; (3) Slightly disagree; (4) Neither agree not disagree; (5) Slightly agree; (6) Agree, or; (7) Strongly agree"). A raw score was created ranging from 7 (minimum) to 56 (maximum) by summing the scores on these eight items ($\Omega = 0.893$). This raw score was then transformed into a percentage score where 0 indicated the lowest possible summed score for the raw items (i.e., a score of 7) while 100 indicated the highest possible score (i.e., a score of 56).

Study 3 measured daily well-being in the evening diary using the World Health Organization- Five Well-Being Index (WHO-5) [100], which contains five items assessing aspects of well-being including emotional and physical vitality. Item timeframe was modified so that participants reported on their experience that day (e.g., "Today, I felt cheerful and in good spirits: At no time (0), some of the time (1), less than half of the time (2), more than half the time (3), most of the time (4), all of the time (5)"). A raw score was created ranging from 0 (minimum) to 25 (maximum) by summing the scores on these 5 items ($\Omega = 0.759$), before being transformed into a percentage score ranging from 0 to 100 following standard scoring instructions.

 

## Analyses

Each dataset was analysed separately but designed to be as parallel in nature as possible, utilising regression or multi-level regression modelling to test whether the four health behaviors significantly predicted well-being after adjusting for covariates. Study 1 used standard linear regression because the data were cross-sectional in nature. Study 2 and 3 used multilevel linear regression to account for the nested structure of the repeated measures daily diary data. For these multi-level models, two variables were created for each of the four self-reported health behaviors (sleep quality, physical activity, total FV consumption, and UPF consumption): a between-person centered variable and a within-person centered variable. The between-person variable was created by averaging a participant's responses across all observations (days) to create a single score per individual, before centring this score around the sample grand mean. As such, this between-person variable represents the individual's deviation from the "average" individual (e.g., Individual 1 averages 10 minutes more physical activity than the average in the dataset). The within-person variable was created by subtracting the participant's own average from each of their observations to create a deviation score for each day. This deviation score reflects the deviation an individual makes on a given day from their own average (e.g., Individual 1 averages 10 minutes more physical activity on Day 1 than their usual). Analyses were conducted in Stata 18 Standard Edition (SE).

For all three datasets, a main effects model was run using the following predictors: (i) covariates (age, gender, ethnicity, SES, BMI, depressive symptoms); and (ii) the linear and quadratic terms for the four health behaviors (sleep quality, physical activity, FV consumption, UPF consumption) at the between-person levels and within-person levels (Studies 2 and 3). Well-being served as the outcome variable in all three datasets. Sleep quantity was also modelled as a predictor for completeness. Quadratic terms were generated after initial exploratory graphs of the variables with lowess curves showed that some variables had non-linear (quadratic) relationships with well-being. Non-significant quadratic terms were then removed if their removal did not significantly reduce the model's explanatory power, examined using the Akaike Information Criteria (AIC) mathematical method, and a post-estimate Wald test on the variable returned a non-significant test score ($p > 0.05$) following a stepwise removal process to model building. Once all non-significant quadratic terms were removed, the resulting model was retained as the final main effects model. Significant associations between the health behaviors and well-being were visualised in Stata to show the between-person and within-person patterns. The user-made command "mlmr2" was used to output R-squared values for the main effect models [101].

Next, interaction models were created by generating all possible two-way interaction terms for the health behaviors including same-level and cross-level interactions and adding them to the final main effects models. As such, all lower order terms from the main effects models were retained. The stepwise removal process was then repeated to remove non-significant interaction terms that did not significantly reduce the model's explanatory power, again assessed using the Akaike Information Criteria (AIC) and a post estimate Wald test. Once all non-significant interaction terms were removed, the resulting model was examined as the final interactions model. The three interactions tested were: (i) between x between person interactions (e.g., "Does someone with high average sleep quality benefit more or less depending on their average FV consumption?"; (ii) between x within person interactions (e.g., "Does someone with high average sleep quality benefit more on days when they eat more FV than normal?" and; (iii) within x within person interactions (e.g., "Does someone benefit more from sleeping better last night if they also ate more FVs that day?"). To mitigate the risk of interpreting interactions that may instead reflect in-sample noise, only those that replicated across at least two datasets were interpreted.

Where appropriate, missing data was treated by either: (i) mean imputation for missing data at item level, or; (ii) using regression with a proxy predictor to generate predicted values as replacements. All variables were centered where possible prior to entry into the model. Categorical predictors were entered using dummy codes. For Study 3's analyses, the final model of each stage was then repeated with the Fitbit measure of physical activity replacing the self-report measure to examine whether similar inferences emerged when using device-measured activity.

The main effects were interpreted using significance levels of 0.05. No corrections for multiple hypothesis testing (e.g., Bonferroni correction) were made, as these corrections assume statistical independence between the tests which is overly conservative for the current analyses. No power analyses were conducted given the datasets had already been collected. Analyses and hypotheses were not preregistered, but the data and code are available open access on the Open Science Framework: https://osf.io/gs8y7/.

## Results

### Descriptive statistics

Table 1 shows the participant characteristics for the three samples. Participants ranged in age from 17 to 25 years old, with the mean sample age of 19–22 years old. Study 1 had the most equal gender balance with 55% women and 44% men, whereas Study 2 and Study 3 had disproportionately more women (72% and 75% women, respectively). A majority of participants were NZ European or European ancestry (75% to 80%), with 20% to 25% diverse ethnicities (Asian, Māori, and Indian being most common). On average, participants reported mid-range childhood socioeconomic status, were within the normal BMI range, and showed slight to mild signs of depressive symptoms.

Table 2 shows the health behaviors and well-being characteristics for three samples at the between-person level. On average, participants slept for 7.3 to 7.6 hours a night, and reported slight issues regarding their sleep quality; exercised for between 26–44 minutes per day; ate between 2.5 to 5 servings of FVs and 1–2 servings of UPFs per day and reported average to moderate well-being as indicated by responses of 48% to 70% on the percentage well-being measures. The

**Table 1. Descriptive statistics for categorical and continuous participant characteristics and covariates from the three studies.**

|  | Study 1: Lifestyle of Young Adults (2021) | Study 2: Daily Life Study (2013–2014) | Study 3: Fitbit Study (2021–2022) |
|---|---|---|---|
| **Categorical Variables** | N (% of 1,032) | N (% of 818) | N (% of 227) |
| Gender |  |  |  |
| *Male* | 450 (43.6%) | 227 (27.8%) | 54 (23.8%) |
| *Female* | 571 (55.3%) | 591 (72.2%) | 169 (74.4%) |
| *Gender Diverse* | 11 (1.1%) | – | 4 (1.8%) |
| Ethnicity |  |  |  |
| *NZ European/European* | 778 (75.4%) | 634 (77.5%) | 182 (80.2%) |
| *Māori* | 30 (2.9%) | 45 (5.5%) | 3 (1.3%) |
| *Asian* | 86 (8.3%) | 87 (10.6%) | 11 (4.9%) |
| *Indian* | 17 (1.7%) | 30 (3.7%) | 6 (2.6%) |
| *Other* | 121 (11.7%) | 22 (2.7%) | 25 (11.0%) |
| **Continuous Variables** | Mean (SD) | Mean (SD) | Mean (SD) |
| Age | 22.1 (2.4) | 19.7 (1.7) | 19.4 (1.2) |
| BMI | 23.8 (5.4) | 24.0 (4.4) | 23.2 (3.8) |
| Childhood SES | 4.7 (1.4) | 4.9 (1.4) | 5.0 (1.3) |
| Depressive symptoms (BP) |  |  |  |
| *Raw scores* | 19.7 (8.1) | 20.3 (6.0) | 17.1 (6.9) |
| *T scores, BP* | 57.3 (9.2) | – | 54.6 (8.1) |
| Depressed mood (WP) | – | Mean[1] ($SD_b$, $SD_w$) | Mean[1] ($SD_b$, $SD_w$) |
|  | – | 4.7 (1.6, 1.4) | 1.51 (1.5, 1.5) |

BMI = Body Mass Index; BP = between-person measure; NZ = New Zealand; SD = standard deviation; $SD_b$ = standard deviation between-person (standard deviation of aggregated mean scores); $SD_w$ = standard deviation within-person (standard deviation of disaggregated scores around individual means, then averaged); SES = socioeconomic status; WP = within-person measure. [1] Mean reflects grand mean.

health behaviors were inconsistently or modestly correlated with one another (S1 Table in S1 File). Descriptively, the strongest correlations were observed between physical activity and FV intake at the between-person level (rs 0.25 to 0.42, $p < .001$) whereas the weakest correlations were observed between UPF consumption and the other health behaviors. The health behaviors were also modestly correlated with several demographic covariates (S2 Table in S1 File). Higher childhood SES predicted better sleep quality and greater FV consumption, whereas female gender predicted less physical activity (both self-report and device-measured) and greater FV consumption, compared to male gender. Across all three studies, better sleep quality consistently predicted lower depressive symptoms.

The intra-class correlation coefficients (ICC) for time-varying measures were mostly below 0.5 indicating more within-person variance than between-person variance (S3 Table in S1 File). The lowest ICCs were observed for sleep quality (ICCs = 0.255 & 0.292) and device-measured physical activity (ICC = 0.282). The highest ICCs were observed for FV consumption (ICCs = 0.503 & 0.567) which reflected a more equal partition of variance at the between- and within-person levels. The ICC for the daily well-being measures were also below 0.5 (ICCs = 0.481 & 0.293) indicating the suitability of within-person analyses.

Table 3 shows the bivariate correlations between the health behaviors and well-being unadjusted for any covariates or considering any shared variance among the health behaviors. Notably, sleep quality showed the strongest correlations with well-being, particularly at the between-person level (rs ~ 0.3–0.5). FV consumption also correlated with well-being both levels in Study 1 and 2, whereas physical activity was less consistently associated with well-being.

## Linear mixed models: Main effects

Table 4 contains the output for the models examining the four health behaviors as simultaneous predictors of well-being, adjusting for the demographic covariates and depressive symptoms. In summary, each of the health behaviors showed

**Table 2. Descriptive statistics for average health behavior engagement and well-being scores across the three studies.**

| Measure | Study 1: Lifestyle of Young Adults (2021; N = 1,021) | Study 2: Daily Life Study (2013–2014; N = 818) | Study 3: Fitbit Study (2021–2022; N = 236) |
|---|---|---|---|
| **Health Behaviors** | Mean (SD) | Mean¹ (SD_b, SD_w) | Mean¹ (SD_b, SD_w) |
| Sleep quantity (hours/night, wins) | 7.25 (2.69) | 7.52 (0.81, 1.22) | 7.60 (0.97, 1.05) |
| Sleep quality (raw scores) | 22.22 (7.18) | 1.46 (0.67, 0.97) | 9.47 (2.19, 2.69) |
| Sleep quality (T scores) | 53.94 (8.28) | – | 48.62 (5.04, 6.03) |
| Sleep quality (T scores, inversed) | −53.94 (8.28) | – | −48.62 (5.04, 6.03) |
| Physical activity (mins/day) | 25.55 (30.25) | 30.60 (23.90, 29.17) | 44.34 (38.79, 32.35) |
| | 15.00 (31.34) | 15.00 (50.00) | 30.00 (60.00) |
| Physical activity (mins/day, F) | – | – | 54.29 (37.45, 45.25) |
| | | | 36.00 (66.00) |
| FV consumption (servs/day) | 4.90 (2.56) | 2.54 (1.37, 1.22) | 3.31 (1.78, 1.30) |
| UPF consumption (servs/day) | 1.70 (0.920) | 1.16 (0.82, 0.97) | 1.32 (0.90, 0.83) |
| **Outcome** | Mean (SD) | Mean (SD_b, SD_w) | Mean (SD_b, SD_w) |
| Well-being (raw scores) | 41.58 (8.44) | 37.58 (9.03, 6.04) | 6.38 (3.41, 4.50) |
| Well-being (% scores) | 69.96 (17.57) | 61.62 (18.80, 12.58) | 49.44 (18.49, 12.38) |

F = Fitbit measured; FV = fruits and vegetables; mins = minutes; SD = standard deviation; $SD_b$ = standard deviation between-person (standard deviation of aggregated mean scores); $SD_w$ = standard deviation within-person (standard deviation of disaggregated scores around individual means, then averaged); servs = servings; UPF = ultra-processed foods; wins = winsorized; For physical activity, second rows indicate median (interquartile range); Well-being measured using the trait Flourishing Scale (Study 1), state Flourishing Scale (Study 2), or state WHO-5 Well-being scale (Study 3). Sleep quality values have been inversed for Study 1 and 3 so that higher values indicate greater sleep quality (T scores, inversed used in all analyses). ¹ Mean reflects grand mean.

**Table 3. Bivariate correlations between health behaviors and well-being.**

| Variables | Between-person | Within-person |
|---|---|---|
| Study 1 (N = 1,032) | | |
| (1) Sleep quality, raw | 0.347*** | – |
| (2) Sleep quality, T scores | 0.348*** | |
| (3) Physical activity | 0.183*** | – |
| (4) FV consumption | 0.259*** | – |
| (5) UPF consumption | −0.040 | – |
| Study 2 (N = 818) | | |
| (1) Sleep quality | 0.498*** | 0.163*** |
| (2) Physical activity | 0.114** | 0.093*** |
| (3) FV consumption | 0.219*** | 0.053*** |
| (4) UPF consumption | −0.165*** | 0.012 |
| Study 3 (N = 236) | | |
| (1) Sleep quality, raw | 0.322*** | 0.122*** |
| (2) Sleep quality, T scores | 0.330*** | 0.129*** |
| (3) Physical activity | 0.054 | 0.069* |
| (4) Physical activity (F) | 0.093 | 0.118*** |
| (5) FV consumption | 0.112 | 0.046 |
| (6) UPF consumption | −0.006 | 0.008 |

*$p < .05$, **$p < .01$, ***$p < .001$; F = Fitbit measured; FV = overall fruit and vegetable consumption; UPF consumption = ultra-processed foods. Correlations are between health behaviors and raw well-being scores. Sleep quality values have been inversed for Study 1 and 3 so that higher values indicate better sleep quality.

significant associations with well-being across the three studies, with higher sleep quality standing out as the most consistent significant predictor of greater well-being at both the between-person and within-person levels (all coefficients significant at $p \leq .001$). More fruit and vegetable (FV) intake and, less consistently, more physical activity, also predicted greater well-being across the three studies, with patterns more consistent at the within-person level. Ultra processed food (UPF) consumption was not consistently related to well-being. The associations between each health behavior and well-being are visualized in . These figures show several isolated curvilinear associations between health behaviors and well-being. The most consistent curvilinear patterns were observed for between-person fruit and vegetable intake and well-being (Fig 3), which showed in Study 1 and 2 that well-being peaked for individuals who ate 5–6 servings per day (~2–3 more servings of FV than the average individual who consumed 2.5 servings/day). For within-person fruit and vegetable intake and well-being, Study 2 showed a curvilinear association whereby well-being peaked around 2 servings more than usual, although in Study 3, the within-person association was strictly linear.

S4 Table in S1 File shows the results from Study 3 when re-running analyses using the Fitbit measure of physical activity. Inferences remained similar, including the significant within-person relationship between physical activity and daily well-being. Thus, regardless of whether physical activity was self-reported or measured by a device, participants reported greater well-being on days when they were more physically active than normal.

### Linear mixed models: Two-way interactions

S5 Table in S1 File shows the full output for the interaction models that contain all terms from the previous main effects models (e.g., covariates, between-person and within-person terms for each health behavior) as well as all significant two-way interactions retained in the models. No replicable two-way interactions were found among the between-person health

**Table 4. Results using health behaviors to predict well-being at the between-person and within-person levels, adjusting for covariates.**

| | Study 1: Lifestyle of Young Adults (2021) | Study 2: Daily Life Study (2013–2014) | Study 3: Fitbit Study (2021–2022) |
|---|---|---|---|
| Outcome Well-being | Flourishing (0–100%) | Flourishing (0–100%) | WHO-5 (0–100%) |
| n (participants) | 1,032 | 817 | 211 |
| k (responses) | 1,032 (1 per person) | 8,994 (~11.0 per person) | 1,076 (~5.1 per person) |
| Design | Cross-sectional | Daily diary | Daily diary |
| **Covariates** | $b$ (SE), $p$ | $b$ (SE), $p$ | $b$ (SE), $p$ |
| Constant | 70.262 (1.13), <0.001 | 61.926 (0.963), <0.001 | 53.461 (2.396), <0.001 |
| Age | −0.032 (0.188), 0.864 | −0.12 (0.233), 0.607 | −0.446 (0.819), 0.586 |
| Gender | | | |
| Male | (reference) | (reference) | (reference) |
| Female | 4.19 (0.876), <0.001 | 2.369 (0.903), 0.009 | −1.995 (2.328), 0.392 |
| Gender diverse | 0.417 (4.128), 0.920 | – | −19.526 (9.889), 0.048 |
| Ethnicity | | | |
| *NZ European* | (reference) | (reference) | (reference) |
| Māori | 3.183 (2.531), 0.209 | 2.105 (1.769), 0.234 | 9.032 (10.041), 0.368 |
| Asian | −3.015 (1.563), 0.054 | −3.887 (1.328), 0.003 | −5.979 (4.315), 0.166 |
| Indian | −0.025 (3.303), 0.994 | −4.746 (2.101), 0.024 | 0.284 (6.817), 0.967 |
| Other | −0.476 (1.324), 0.719 | 0.742 (2.485), 0.765 | −3.408 (3.108), 0.273 |
| BMI | −0.054 (0.079), 0.497 | −0.297 (0.093), 0.001 | 0.308 (0.272), 0.258 |
| Childhood SES | 2.032 (0.331), <0.001 | 0.35 (0.297), 0.238 | 1.712 (0.846), 0.043 |
| Depressive symptoms | | | |
| *Between-person* | −0.943 (0.054), <0.001 | −0.421 (0.069), <0.001 | −0.647 (0.132), <0.001 |
| *Within-person proxy* | – | −3.443 (0.079), <0.001 | −3.581 (0.226), <0.001 |
| $R2_w$, increase | – | 0.095, 0.095 | 0.066, 0.066 |
| $R2_b$, increase | 0.377, 0.377 | 0.070, 0.070 | 0.247, 0.247 |
| Total R2, increase | 0.377, 0.377 | 0.165, 0.165 | 0.313, 0.313 |
| **Health Behaviors** | | | |
| *Between-person* | | | |
| Sleep quantity | 0.279 (0.158), 0.078 | −0.312 (0.485), 0.520 | −0.8 (0.898), 0.373 |
| Sleep quantity (quad) | – | −0.873 (0.345), 0.011 | – |
| Sleep quality | 0.185 (0.06), 0.002 | 8.587 (0.638), <0.001 | 1.112 (0.236), <0.001 |
| Sleep quality (quad) | −0.009 (0.004), 0.033 | – | – |
| Physical activity | 0.115 (0.025), <0.001 | 0.005 (0.018), 0.77 | 0.044 (0.027), 0.099 |
| Physical activity (quad) | −0.001 (0.000), 0.035 | – | – |
| FV consumption | 1.038 (0.204), <0.001 | 1.37 (0.338), <0.001 | 1.084 (0.594), 0.068 |
| FV consumption (quad) | −0.162 (0.052), 0.002 | −0.38 (0.159), 0.017 | – |
| UPF consumption | 1.022 (0.512), 0.046 | −1.284 (0.500), 0.01 | −1.072 (1.115), 0.336 |
| *Within-person* | | | |
| Sleep quantity | n/a | −0.242 (0.094), 0.01 | 0.408 (0.306), 0.183 |
| Sleep quality | n/a | 2.355 (0.135), <0.001 | 0.483 (0.066), <0.001 |
| Physical activity | n/a | 0.038 (0.004), <0.001 | 0.04 (0.009), <0.001 |
| FV consumption | n/a | 0.546 (0.102), <0.001 | 0.957 (0.274), <0.001 |
| FV consumption (quad) | n/a | −0.135 (0.058), 0.020 | – |
| UPF consumption | n/a | 0.21 (0.122), 0.085 | 0.402 (0.416), 0.334 |
| $R2_w$, increase | – | 0.117, 0.022 | 0.095, 0.029 |
| $R2_b$, increase | 0.437, 0.060 | 0.185, 0.115 | 0.322, 0.075 |
| Total R2, increase | 0.437, 0.060 | 0.301, 0.136 | 0.417, 0.104 |

*(Continued)*

**Table 4.** (Continued)

|  | Study 1: Lifestyle of Young Adults (2021) | Study 2: Daily Life Study (2013–2014) | Study 3: Fitbit Study (2021–2022) |
|---|---|---|---|
| AIC | 8288.643 | 71702.25 | 8721.84 |
| BIC | 8382.489 | 71886.96 | 8841.384 |

AIC = Aikake's Information Criterion; BIC = Bayesian Information Criterion; FV = overall fruit and vegetable consumption; Quad = quadratic term; R2 = R squared; $R2_w$ = Total R squared explained by the predictors at the within-person level; $R2_b$ = Total R squared explained by the predictors at the between-person level; SES = Socioeconomic status; UPF consumption = ultra-processed foods.

behaviors (between x between) or across levels (between x within). However, one replicable interaction was found among the within-person health behaviors (within x within) which replicated across Study 2 and Study 3. Specifically, sleep quality interacted with FV consumption to predict within-person changes in well-being in Study 2 (Sleep quality x FV consumption quadratic $b(se)$ = 0.108 (0.051), $p$ = 0.034 and linear $b(se)$ = −0.186 (0.104), $p$ = 0.074) and Study 3 (Sleep quality x FV consumption linear $b(se)$ = −0.161 (0.049), $p$ = 0.001). The interactions between sleep quality and fruit and vegetable consumption are visualized in Fig 5. Study 2 suggested a protective association where eating the usual or above the usual amount of FV helped protect people from the negative effects of a poor night's sleep on well-being. Study 3 again suggested a similar association whereby more than usual FV consumption protected people from the detriments of a poor night's sleep. The interaction observed in Study 3 remained significant when using Fitbit-measured physical activity in the full interaction model (S6 Table in S1 File).

## Discussion

Our goal was to investigate how consistently the "big three" health behaviors of sleep quality, physical activity, and diet predicted psychological well-being among young adults. We sought to improve on prior research in three ways: by testing associations with positive well-being not just mental ill-being, by testing health behaviors together in the same datasets to determine their additive or synergistic associations with well-being, and, by disaggregating between-person and within-person associations. Table 5 summarizes seven key insights from the findings.

Across three datasets, sleep quality was the only health behavior that consistently predicted well-being at both the between-person and within-person levels. Specifically: (i) individuals with better sleep quality reported higher well-being than those with poorer sleep quality, and (ii) on days when people slept better than usual, they reported better well-being. FV consumption was nearly as consistent, showing significance in all three datasets except at the between-person level in Study 3 ($p$ = 0.068). The pattern for FV intake indicated that: (i) people who consumed more FVs generally reported higher well-being, and (ii) on days when they ate more FVs, they experienced better well-being. The primacy of these behaviors may stem from their strong, direct effects on various biological processes. For instance, chronic poor sleep disrupts healthy biological functions, including immune system activation [102], cognitive impairment [103], and excessive daytime tiredness, which correlates with increased negative events and reduced positive affect [104,105]. This disruption likely underlies sleep quality's strong link to well-being. Similarly, FV consumption may be significant due to the rapid effects of increased vitamins and minerals on energy levels and subjective vitality—a key aspect of positive mood [106]—which can accumulate within days [41,107].

Physical activity's association with well-being was less consistent at the between-person level. This inconsistency could be due to unexamined moderating factors. Research suggests physical activity is sensitive to moderation by variables such as physique anxiety [108], intrinsic motivation [109], and social influences [110]. For instance, individuals who exercise excessively might do so for reasons that disrupt its positive impact on well-being, such as upward comparisons and unattainable body ideals [111–113]. While physical activity is crucial for health, its impact on well-being at

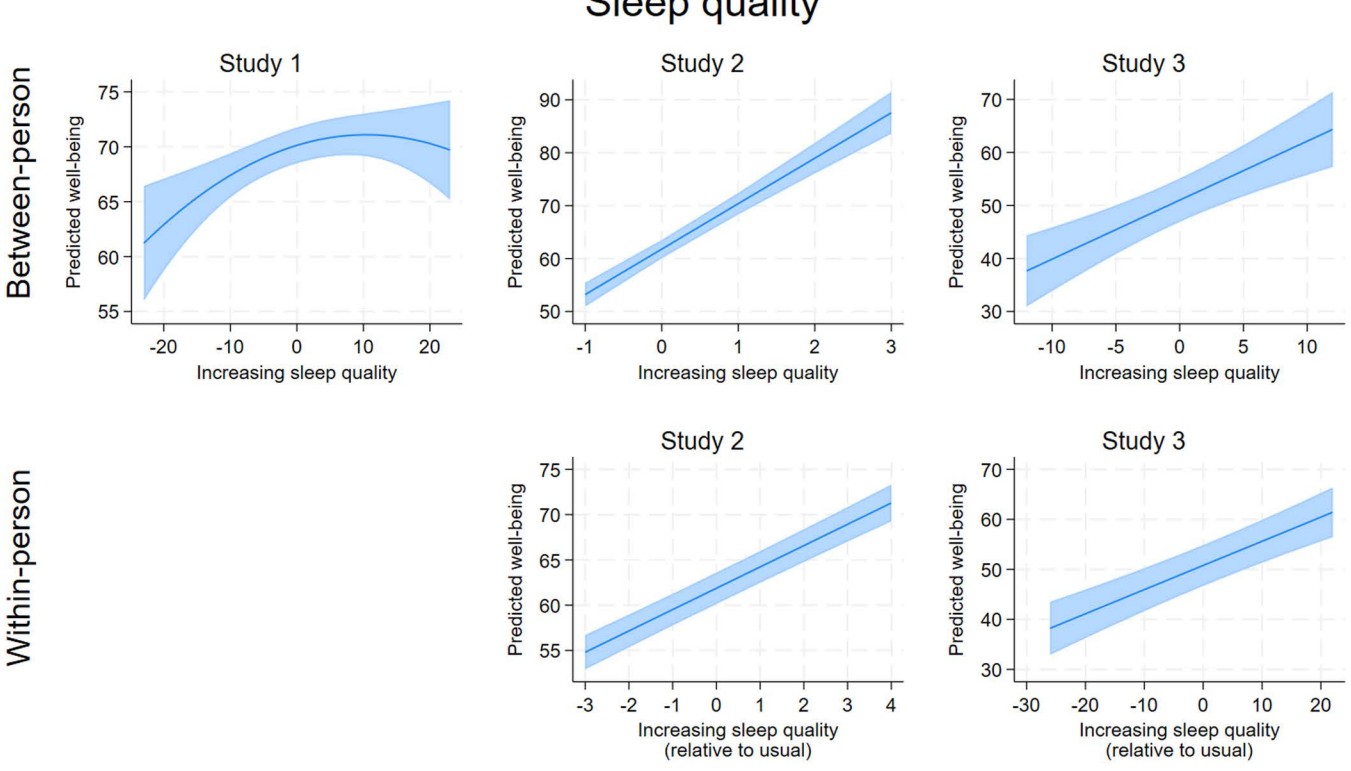

**Fig 1. A collated visualization of the association between sleep quality and predicted well-being at the between-person and within-person levels.** Lines indicate adjusted predictions; shaded areas indicate confidence intervals. Gray areas indicate a non-significant association.

the between-person level may vary based on individual differences and contextual factors, making its association with well-being less consistent than that of sleep quality and FV consumption. At the within-person level however, physical activity was consistently linked to well-being (i.e., on days when people exercised more than usual, they reported better well-being). This finding aligns with previous research showing that moderate to vigorous physical activity offers immediate and lasting mood benefits [37]. Increased physical activity raises arousal and triggers endorphin release during and after exercise [114–116], which may explain its consistent within-person association with well-being. Exercise also fosters feelings of accomplishment and agency, further enhancing well-being [46].

When the self-report models were re-run using Fitbit-measured physical activity, inferences remained largely similar, suggesting that the two are interchangeable when examining how vigorous physical activity associates with well-being in this population. Although this was tested in only one study, these findings are promising for researchers interested in wearables and the correspondence to self-report survey measures. One implication is that researchers without access to wearable devices can have more confidence that self-report measures of vigorous and moderate activity are capturing similar associations in this population, while researchers concerned about participant fatigue from multiple self-reports can consider using Fitbits to mitigate this. We did try to measure sleep metrics including proxy measures of sleep quality from the Fitbits, but the data quality was extremely poor and unreliable [117]. However, as a measure of physical activity, the Fitbits performed well.

UPF intake was not a consistent predictor of well-being. It was linked to higher well-being in Study 1, lower well-being in Study 2, and no differences in Study 3. This inconsistency challenges the extensive literature associating increased

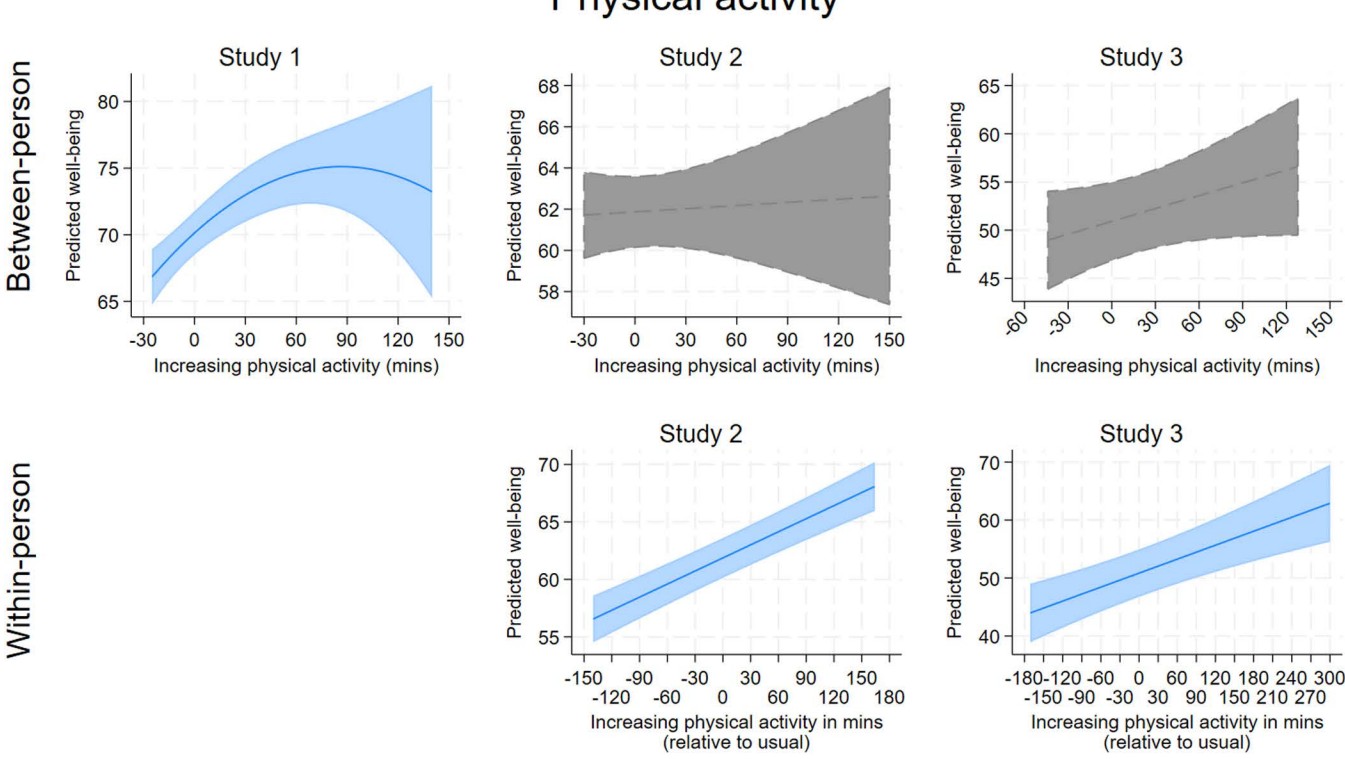

**Fig 2. A collated visualization of the association between physical activity and predicted well-being at the between-person and within-person levels.** Lines indicate adjusted predictions, shaded areas indicate confidence intervals. Gray areas indicate a non-significant association.

UPF consumption with negative health outcomes [118,119,120,121–125]. Although the literature linking UPF with negative mental health outcomes is smaller, a recent meta-analysis found a link between higher UPF intake and increased odds of depressive and/or anxiety symptoms [126]. In our study, small but significant correlations between UPF intake and greater depressive symptoms were found in Studies 1 and 2, but not in Study 3, suggesting that UPF intake may be more relevant to negative mental health outcomes. Given mental illness and well-being are interrelated, it was expected that well-being would show a similar association with UPF intake, but it did not. It is possible that other variables, such as disordered views of unhealthy food, could moderate the relationship between UPF intake and well-being, similar to how 'Fitspiration' social media content has been shown to attenuate the positive association between physical activity and mental well-being [111]. Alternatively, the null patterns observed for UPF could suggest that increased UPF consumption may not inherently be a "bad thing" for well-being among young adults. If they maintain quality sleep, regular exercise, and sufficient FV intake, they might consume UPFs without adverse effects on positive functioning or subjective well-being. However, this does not imply that increasing UPF consumption will have no impact on well-being, nor that these patterns apply to older adults. UPF consumption can still lead to physiological and psychological effects detrimental to other health behaviors. For instance, UPFs are low in fibre and high in sugar, causing blood glucose spikes followed by crashes, which may reduce energy levels and discourage physical activity [127]. These crashes might also lead to short-term mood or well-being changes not evident at the day level. Additionally, UPFs are hyperpalatable [128] and may displace less appetizing but healthier food categories like FVs.

For the most part, the relationships between the big three health behaviors and well-being were additive, with each independently contributing to well-being. The modest correlations between these behaviors in young adults suggest they

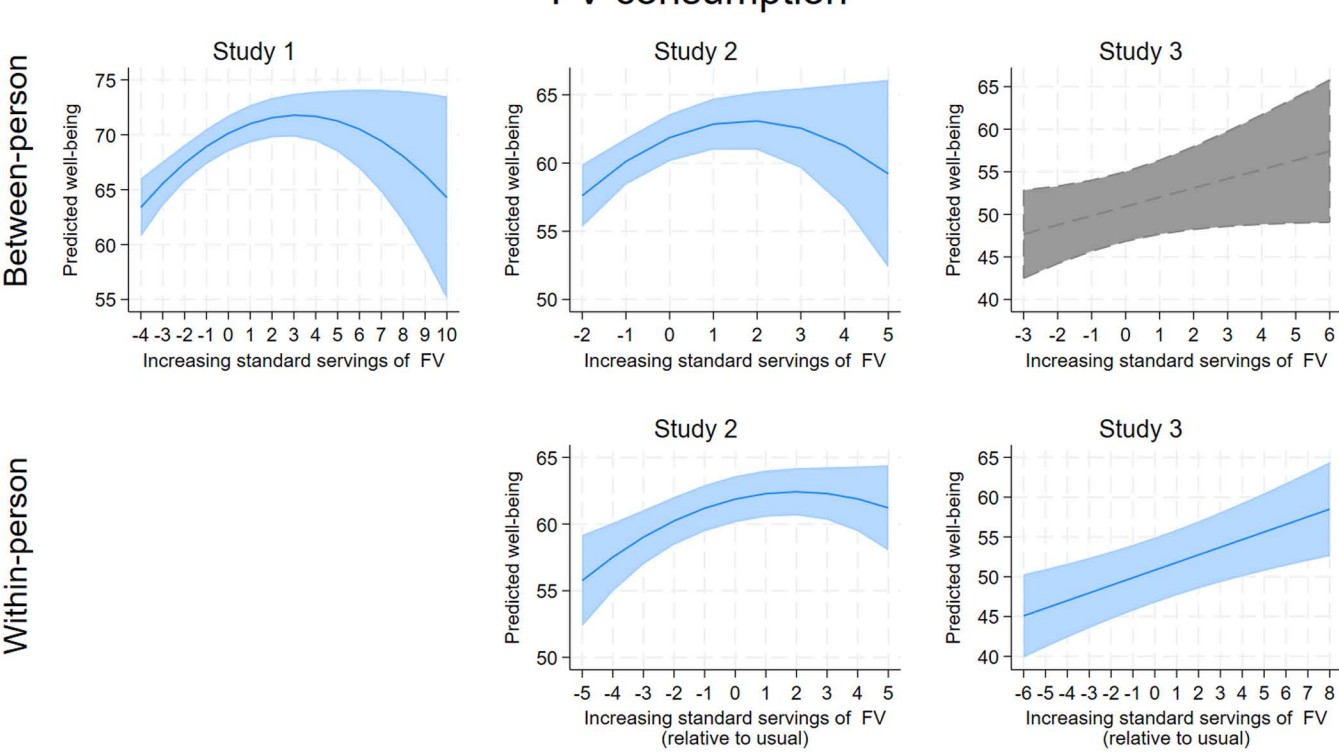

**Fig 3. A collated visualization of the association between fruit and vegetable (FV) consumption and predicted well-being at the between-person and within-person levels.** Lines indicate adjusted predictions, shaded areas indicate confidence intervals. Gray areas indicate a non-significant association.

do not cluster together, and each may uniquely contribute to well-being without diminishing returns or requiring a synergistic threshold for effectiveness. However, one interaction emerged in the two daily diary datasets: the interaction between sleep quality and FV consumption at the within-person level. This consistent pattern showed that increased FV consumption mitigated the impact of a poor night's sleep, while FV was less crucial to well-being following good sleep. The replication of this interaction across datasets suggests this association may be "real". One explanation is that poor sleep typically lowers energy and mood [43], and increased FV intake could restore nutrients needed to improve energy and reduce fatigue. FV consumption is linked to immediate increases in vitamins and minerals [41,107], which enhance energy levels and subjective vitality—a key component of positive mood [106]. Thus, increased FV intake may offset the energy deficit caused by poor sleep, boosting vitality and positive affect.

## Implications

The results of this paper suggest that everyday lifestyle habits may be more than basic physiological needs; they may be associated with improved mental health that takes someone from "surviving" to "thriving". Foremost, sleep quality may be a priority health target for interventions to promote well-being. When added to the model first, sleep quality typically explained more of the variance in well-being than the other health behaviors, both at the within-person and the between-person level (see S7 Table in S1 File). Some caveats apply, however, because sleep quality may not be the most practical intervention target. Improving sleep is challenging due to environmental factors like blue-light exposure and caffeine, which disrupt sleep [129,130]. Additionally, sleep is difficult to influence directly, as it occurs while individuals are

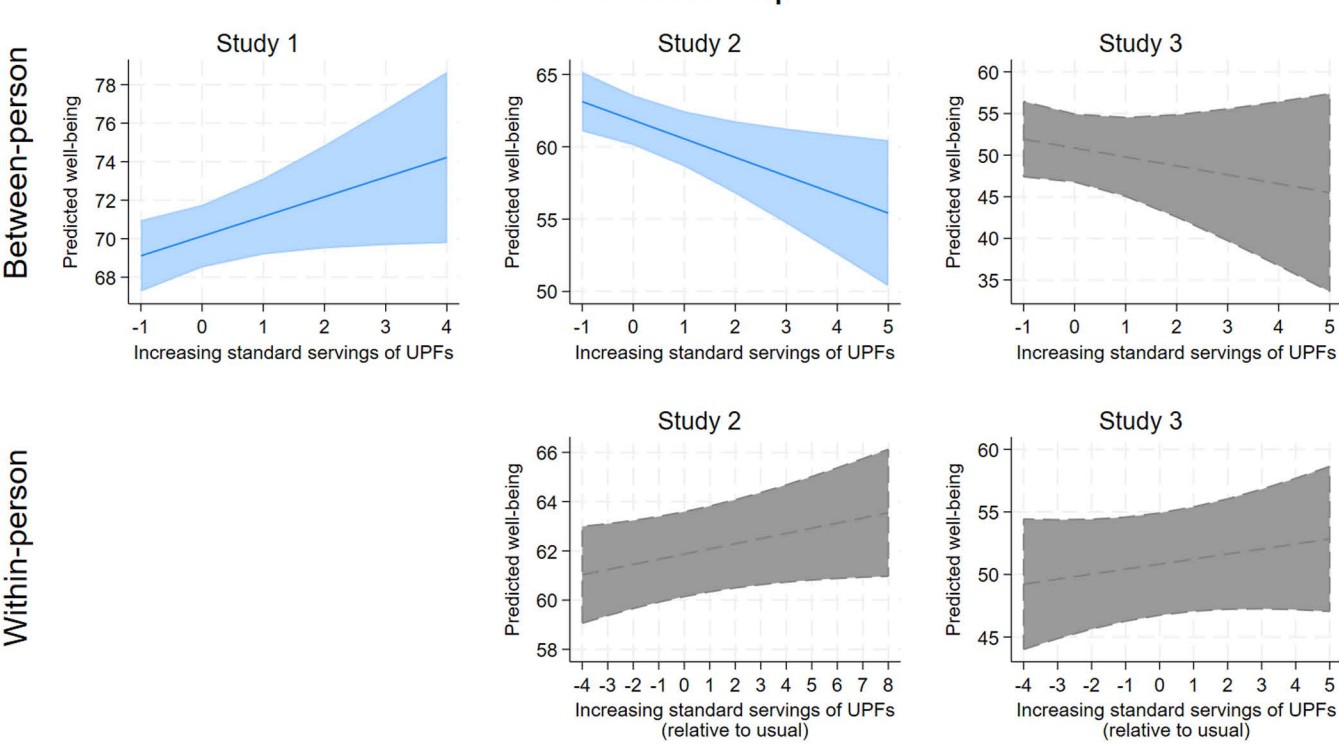

**Fig 4. A collated visualization of the association between ultra-processed food (UPF) consumption and predicted well-being at the between-person and within-person levels.** Lines indicate adjusted predictions, shaded areas indicate confidence intervals. Gray areas indicate a non-significant association.

unconscious, so only indirect measures, such as reducing caffeine and blue light, can be taken. Therefore, targeting physical activity and FV consumption may be more effective, as they are easier to influence directly. Interventions could focus on between- and within-person changes, like screening for low physical activity or FV consumption to boost overall levels or encouraging people to be more active and eat more FV than they normally do. There is also a need to address psychological issues, such as body dysmorphia or restrictive eating, which may disrupt the positive effects of health behaviors on well-being [111]. Lastly, interventions solely aimed at reducing UPF consumption may not consistently improve well-being. A holistic approach that considers both dietary choices and other health behaviors is more likely to yield improvements.

Secondly, the observed two-way interaction between sleep quality and FV consumption may further optimize interventions to improve well-being. It is exciting to observe that increasing FV intake the day after poor sleep could potentially mitigate its negative effects. Future research should focus on adaptive interventions that recommend FV consumption following poor sleep or use randomized designs to test causal impacts. Since young adults often report poor sleep [131], promoting increased FV intake could be a practical strategy to counteract the mental effects of poor sleep, pending replication in intervention studies.

Finally, the unique, additive associations of each of the big three health behaviors with well-being suggest that young adults can achieve noticeable improvements by focusing on any one behavior, rather than making comprehensive lifestyle changes. This means individuals have various options to enhance their well-being. For example, someone with poor sleep hygiene due to external factors could improve their well-being through diet or exercise, while those averse to physical activity could benefit from increasing FV intake. Intervention research should tailor support to individual circumstances,

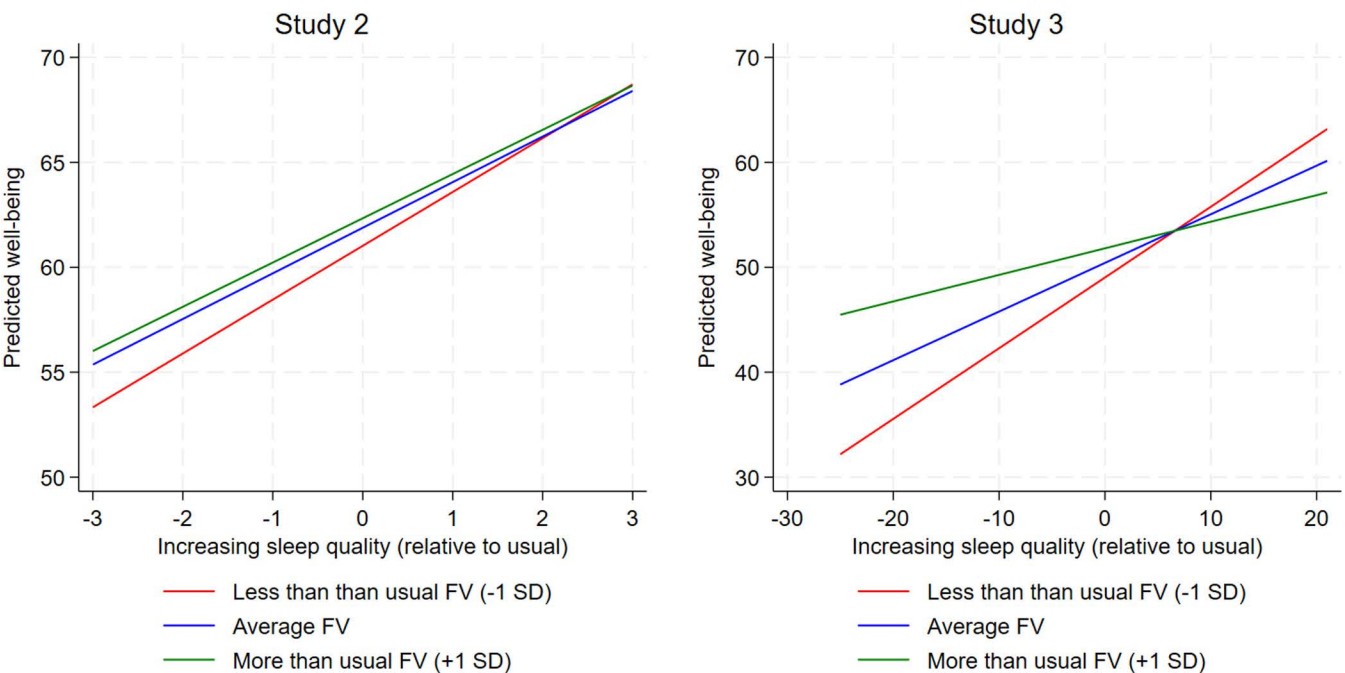

**Fig 5. A graph visualizing the within-person interaction between sleep quality and fruit and vegetable (FV) consumption in predicting well-being observed in Study 2 and Study 3.** SD=within-person standard deviation.

**Table 5. Seven insights from synthesising across the three datasets.**

| Seven insights regarding the big three health behaviors and well-being. |
| --- |
| 1. Sleep quality was the most consistent predictor of well-being, followed by fruit and vegetable consumption and then physical activity. |
| 2. Associations were found at the between-person and within-person levels, meaning:<br>a) People who slept better and/or ate more fruits and vegetables reported feeling better than individuals who did not. However, this same pattern was less consistently observed for physical activity.<br>b) On days when people slept better, exercised more, and/or ate more fruits and vegetables than they normally did, they reported feeling better than on days they did less of these health behaviors. |
| 3. Consumption of processed foods was not consistently linked to lower well-being, casting doubt on its relevance to well-being. However, greater consumption of processed foods was related to higher scores on the depressive symptoms covariate in two out of three datasets tested. |
| 4. The associations between health behaviors and well-being persisted despite controlling for key demographics and depressive symptoms, suggesting their unique contribution to positive mental states. |
| 5. The associations between health behaviors and well-being were largely additive, suggesting young adults can see meaningful improvement via each and all of the health behaviors. |
| 6. There was evidence for a protective interaction between sleep quality and fruit and vegetable consumption, where eating more fruits and vegetables may help protect from a poor night's sleep. |
| 7. Self-reported physical activity and Fitbit-recorded physical activity showed similar associations with well-being, suggesting that the two approaches are interchangeable for research examining this association. |

preferences, and personalities. Given young adults' limited time and resources [22,24], targeting single health behaviors may be more effective than overwhelming them with multiple changes. This contrasts with findings in older populations, where a holistic approach to multiple healthy behaviors is more predictive of positive health outcomes [132,133]. Thus, interventions for young adults should balance between addressing immediate needs for results (i.e., focusing on one health behavior) and fostering long-term health outcomes (i.e., promoting multiple health behaviors together).

## Strengths and limitations

Strengths of the paper included its methodological and statistical approach. The analyses adopted a 'bottom-up' approach, focusing solely on what the data revealed across three datasets [134]. This approach ensured that inferences were grounded in the sample context (young adults) and that replicability and association patterns across studies guided interpretation. The analyses used complex multilevel modelling to account for the nested nature of two of the three data-sets and unearth rich detail at different levels of analysis. Partitioning between- and within-person components of each association allowed for different inferences regarding timeframes. Additionally, examining the health behaviors simultane-ously allowed for each association to be discussed in the context of "the real world" where these behaviors occur together, rather than in a vacuum where only one is acknowledged at a time. Finally, a wide variety of covariates were controlled for, including depressive symptoms, which supports these associations between the health behaviors and well-being are not simply due to covariation with mental ill-health or demographic characteristics. Of course, there may be other unac-counted variables that could be driving these associations, which is a limitation of all observational research.

This paper had several limitations. First, the observational nature of the data limits causal inferences about health behaviors and well-being; particularly for Study 1, this means we cannot rule out the possibility of reverse causality (where well-being may be predicting health behavior engagement). For Studies 2 and 3, temporal precedence is present given well-being was assessed at the end of each day, while some of the health behaviors were assessed earlier in the day (e.g., sleep quality assessed in the morning and predicting well-being later in the day). This paper lays the groundwork for future research to investigate these associations at a higher level of temporal granularity, such as repeated measurements within the same day.

Second, the use of unstandardized regression coefficients due to the multilevel data structure limits direct comparison of effect sizes across health behaviors. However: (i) comparing the R squared increase for each health behavior when added first to the null model(s) allows us to compare the strength of their associations with well-being (S7 Table in S1 File), and; (ii) the unstandardized coefficients provide insights framed in real-world contexts, which can inform individuals and intervention efforts in a more practical way than standardized coefficients. For example, Study 3's unstandardized coefficients suggest that an individual could expect the same gain from eating an extra apple that day as they would from exercising 25 minutes more, which may be an easier way of understanding these associations in practical terms.

Third, the lack of power analyses, given the datasets were already collected, introduces the possibility of type-2 errors, especially for non-significant associations with physical activity and UPF consumption. However, literature suggests that these associations may be moderated by variables not examined here, such as intrinsic motivation [135]. The non-significant findings could stem from unexamined moderators rather than low power, with narrow confidence intervals for significant associations supporting the robustness of the results. Lastly, the study focused on sleep, physical activity, and diet, excluding other health behaviors like smoking, vaping, drug use, and alcohol consumption, which could also affect well-being [136]. Future research should investigate a broader range of health behaviors to fully understand their impact on well-being.

Fourth, not every measure used to assess participants was the same between studies, due to the studies being designed at different times and originally for different analyses. For example, Study 1 used the 8-item variation of the PROMIS Sleep Disturbance Short Form to assess sleep quality, while Study 3 used the 4-item variation. As such, observed differences in associations from study to study may be in part due to these different measures. Future studies

should consider re-using the same measures from only one of this paper's studies to better confirm or contrast the associations presented.

Finally, data for both Study 1 and Study 3 were collected during 2021−2022, a time where the COVID-19 pandemic was still prevalent and disruptive to daily routines. As such, some of the findings presented in this paper may be best understood in the context of this time period. For example, UPF's significant between-person association with well-being observed in Study 1 may be due to individuals worsening their diet's quality during the pandemic because of social anxiety, anxiety and distress. However, the health behaviours between the studies were more similar than different, indicating that the patterns observed may be somewhat stable irrespective of the COVID-19 pandemic. With the COVID-19 pandemic no longer affecting the majority of the world's daily life, future research may allow for these findings to be either be generalized or situated firmly within this past context.

## Conclusion

In summary, this paper aimed to further understand how the "big three" health behaviors of sleep, physical activity, and dietary factors associate with well-being. It did so by examining all three health behaviors as simultaneous predictors of well-being at both the between-person and within-person levels, and observing a consistent two-way interaction that may hold benefit for intervention efforts. Finally, these associations were examined after controlling for a wide variety of covariates, including depressive symptoms, suggesting these positive associations are robust and meaningful. Overall, this paper provides evidence that lifestyle choices are promising avenues for potential intervention to help young adults thrive, not just survive.

## Supporting information

**S1 File.** S1 Table. Bivariate correlations among the health behaviors. *p < .05, **p < .01, ***p < .001; F = Fitbit measured; FV = fruit and vegetable consumption; UPF consumption = ultra-processed foods (sweets and chips combined); Correlations above diagonal = between-person, below diagonal = within-person. Sleep quality measures are those used in each study's respective final model (i.e., Study 1 and 3 = T scores, Study 2 = raw scores). Sleep quality values have been inversed for Study 1 and 3 so that higher values indicate greater sleep quality. **S2 Table.** Bivariate correlations between covariates and health behaviors (between-person). *p < .05, **p < .01, ***p < .001; BMI = body mass index; F = Fitbit measured; Raw = raw sleep quality scores; SES = socioeconomic status; T = sleep quality T scores. [1] male (0) vs. female/gender diverse (1). [2] NZ European (0) vs non-NZ-European/mixed ethnicity (1). [3] raw scores on depressive measures. Sleep quality values have been inversed for Study 1 and 3 so that higher values indicate greater sleep quality. **S3 Table.** Intra-class coefficient (ICC) values for the health behaviors and well-being measures in the daily diary studies. FV = fruit and vegetables; ICC = intra-class correlation coefficient reflecting the percentage of variance attributable to between-person differences; SE = standard error; UPF = ultra-processed foods. The ICC represents the variance attributable to between-person differences (ICCs > .5 indicate more variance is between people; ICCs < .5 indicate more variance is within-people). Daily well-being was measured using the Flourishing Scale in Study 2 and the WHO-5 Well-Being Scale in Study 3 both adapted to daily reporting. **S4 Table.** A table comparing the outputs from Study 3's models using the self-report measure of physical activity and Fitbit device measured physical activity. AIC = Aikake's Information Criterion; BIC = Bayesian Information Criterion; FV = fruit and vegetable consumption; Quad = quadratic term; SES = socioeconomic status; UPF consumption = ultra-processed foods. **S5 Table.** Results using health behaviors to predict well-being at the between-person and within-person levels, adjusting for covariates and allowing for two-way interactions between the health behaviors. AIC = Aikake's Information Criterion; (b) = between-person; BIC = Bayesian Information Criterion; FV = overall fruit and vegetable consumption; Quad = quadratic term; SES = socioeconomic status; UPF consumption = ultra-processed foods; (w) = within-person. **S6 Table.** A table comparing the outputs from Study 3's two-way interaction models using the self-report measure of physical activity and Fitbit device measured physical activity. AIC = Aikake's

Information Criterion; (b)=between-person; BIC = Bayesian Information Criterion; FV = overall fruit and vegetable consumption; Quad = quadratic term; SES = socioeconomic status; UPF consumption = ultra-processed foods; (w)=within-person.

**S7 Table.** A comparison of each health behavior's impact on the model's R squared value when added first to the null model (covariates only). FV = overall fruit and vegetable consumption; R2 = R squared; $R2_w$ =Total R squared explained by the predictors at the within-person level; $R2_b$ =Total R squared explained by the predictors at the between-person level; SES = Socioeconomic status; UPF consumption = ultra-processed foods. [1] R squared can occasionally decrease when adding predictors to multilevel models due to: (i) chance fluctuation, or (ii) the new predictor being redundant to a predictor already in the model. See Snijders & Bosker (1999; 138) for more detail.

(DOCX)

## Author contributions

**Conceptualization:** Tamlin S. Conner.

**Data curation:** Jack R.H. Cooper.

**Formal analysis:** Jack R.H. Cooper, Robin S. Turner.

**Funding acquisition:** Tamlin S. Conner.

**Methodology:** Tamlin S. Conner.

**Software:** Robin S. Turner, Tamlin S. Conner.

**Validation:** Tamlin S. Conner.

**Visualization:** Jack R.H. Cooper.

**Writing – original draft:** Jack R.H. Cooper.

**Writing – review & editing:** Robin S. Turner, Tamlin S. Conner.

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
