## [Decision Letter · Decision Letter 0]

25 Feb 2025

Dear Dr. Conner,

Thank you for submitting your manuscript to PLOS ONE. After careful consideration, we feel that it has merit but does not fully meet PLOS ONE’s publication criteria as it currently stands. Therefore, we invite you to submit a revised version of the manuscript that addresses the points raised during the review process.

We look forward to receiving your revised manuscript.

Kind regards,

Elma Izze Da Silva Magalhães

Academic Editor

PLOS ONE

Journal Requirements:

2**.** Thank you for uploading your study's underlying data set. Unfortunately, the repository you have noted in your Data Availability statement does not qualify as an acceptable data repository according to PLOS's standards.

3. We note that you have referenced (103). Conner TS, et al. Ultra-processed foods predict poorer mental health in young adults: a cross-sectional investigation in three countries. Unpublished manuscript. University of Otago, New Zealand; 2021.) which has currently not yet been accepted for publication. Please remove this from your References and amend this to state in the body of your manuscript: 103). Conner TS, et al. Ultra-processed foods predict poorer mental health in young adults: a cross-sectional investigation in three countries. Unpublished manuscript. University of Otago, New Zealand; 2021.) as detailed online in our guide for authors

http://journals.plos.org/plosone/s/submission-guidelines#loc-reference-style.

4. We notice that your supplementary [tables] are included in the manuscript file. Please remove them and upload them with the file type 'Supporting Information'. Please ensure that each Supporting Information file has a legend listed in the manuscript after the references list.

Reviewers' comments:

Reviewer's Responses to Questions

**Comments to the Author**

1. Is the manuscript technically sound, and do the data support the conclusions?

Reviewer #1: Yes

Reviewer #2: Yes

2. Has the statistical analysis been performed appropriately and rigorously?

Reviewer #1: Yes

Reviewer #2: Yes

3. Have the authors made all data underlying the findings in their manuscript fully available?

Reviewer #1: Yes

Reviewer #2: Yes

4. Is the manuscript presented in an intelligible fashion and written in standard English?

Reviewer #1: Yes

Reviewer #2: Yes

Reviewer #1: The explanation of the dual continuum models is not fully understood in the manner authors writing in the introduction. It is recommended that the authors re write the explanation in a more readable manner in the introduction section.

The authors could relate that Baltic sea diet as fish consumption in the introduction. Most of the readers are not familiar with this diet. Could they explain a little bit more about that diet before writing the results of the study.

The manuscript doesn’t contain the figures of the article.

The data analysis plan was really carefully done. The authors insisted on between subjects analysis and within subject analysis. But I am afraid the mixed model table is a bit hard to relate to that. I am referring to table 4 which is very condensed and full of information. Could the authors write a footnote or more explanation for their mixed model technique. I think the title of the graphs indicate the answer to some of my questions about the results section. I don’t have the graphs in my pdf sent from the journal.

The authors in discussion section reported that the physical activity was the least effective variable for well being based on their analysis and yet they say their results are aligned with others relating more sleep is linked with better mood. If the authors want to report this finding they should get the results section between good sleep quality and mood in the three data sets before jumping to this conclusion.

Reviewer #2: Decision Letter

Dear Authors,

First of all, I congratulate you on your efforts to study such complex interrelationships as those of behavioral and well-being nature. These relationships involve very deep and interconnected mechanisms, which are still not fully understood in all their complexity. The article addresses topics that, although widely studied separately, still lack this integrated and multifaceted perspective to provide a better understanding. I recognize the value of this work for studying different databases, with extensive analyses and not-so-simple interpretations, but I have some comments and suggestions for improving the presentation of these findings.

Title:

The title is somewhat sensationalist regarding the phrase "From surviving to thriving." I think it could perfectly well be just "How sleep, physical activity, and diet shape well-being in young adults." This is merely a suggestion, not an imposition, but I consider that "From surviving to thriving" feels disconnected from the article as a whole, as this statement is never revisited in the text, and the authors did not discuss in the discussion how well-being could be regained or improved based on the observed results. To include this element in the title, it would have been necessary to address how these lifestyle habits are basic human needs and how improving these factors can positively impact individuals' well-being, taking them from basic needs (surviving) to truly living well (thriving).

Abstract

It is appropriate and well summarizes the main points of the study.

Introduction

The introduction, although providing a good contextualization of the studied topic and good justifications for the study's objective, is unnecessarily lengthy. With so much information, the introduction feels long, almost giving the impression that much is already known about the subject, somewhat diminishing the need for another study on this topic. I suggest reducing the text a bit, removing some details about other studies, and leaving this type of comparison between previous studies more for the discussion.

The section of the introduction that talks about the objectives should conclude the introduction; there is no need to start detailing the datasets in the introduction, as this fits better in the methods section.

The paragraph starting at line 158 also seems somewhat unnecessary, as it tries to justify something already justified earlier about the importance of the "big three behaviors." I understand that this information strengthens the rationale for the choice of adjustment variables, but it does not need to be discussed so early in the article.

Methods

The authors do not mention how participants were selected for the original studies. The lack of this information compromises the understanding of possible limitations and biases in the selection process, issues that may affect the representativeness and generalizability of the obtained results. I suggest that the authors briefly mention the type of method used to select participants for each dataset used, and whether participants were selected differently within the same dataset (convenience sample? random sampling?). Some parts of the methods suggest the use of convenience samples, but this needs to be clearer.

It is mentioned that in Study 1, participants who showed evidence of response bias were excluded. What would these response bias evidences used as criteria be? How was this evaluated and judged to decide who would be excluded for this reason?

The eligibility criteria of the original studies also need to be clarified, not just the reasons for exclusion from the analysis.

In line 251, it is mentioned that 818 individuals aged between 17 and 25 years had usable data. This is confusing, as from the beginning (abstract and parts of the introduction), it is stated that participants were between 18 and 15 years old. If one of the datasets included eligible individuals aged 17, then the authors are indeed including individuals aged 17 to 25 in the total N. This needs to be clear from the beginning of the text, preferably from the abstract.

Results

In Table 1, I suggest presenting only the mean and standard deviation of age in each study. I see no reason to present it in categories, especially considering that some ages have very few or no participants, resulting in too much unused information in the table. I also suggest reviewing the rounding of decimal places, as some items do not add up to 100%.

The way SES was assessed was not clear enough in the methods, making it somewhat difficult to understand what these values in Table 1 mean. How was this assessed? What do the mean and standard deviation in the table represent? I suggest explaining this better in the methods section or adding a footnote to facilitate interpretation in the table.

In Table 2, I suggest removing the term "daily" from the title, as Study 1 does not use daily information.

As a general critique of all presented tables, the sample N used in the analyses should always appear, either in the table title, footnote, or within the table itself, as besides the differences in N between studies, it is necessary to be transparent about whether any data remained missing. The article is quite long, and it is difficult to keep searching for this information in other parts of the article to recall.

Another important point: was presenting median and interquartile range considered for variables with skewed distributions? I ask because certainly some of the studied variables have a skewed distribution, and using the mean may not be ideal as it is influenced by extreme values. I suggest evaluating this point.

Discussion

Data from 2021 were used in Study 1. How might the results have been affected by the pandemic context, given that in 2021 the situation was still quite serious, and people had not yet fully resumed their routines and habits, especially regarding the three main behaviors studied? Study 3 also used data collected during a period when COVID-19 was still a major concern. Was this considered? It seems that the results obtained in Study 1, which observed higher consumption of ultra-processed foods, might be related to this data collection period, especially considering the observed correlation between this consumption and depressive symptoms. In many cases, individuals worsened their diet quality during the pandemic due to social isolation, anxiety, and distress during that period. It is possible that many people found some comfort in consuming these unhealthy foods, which might have brought some well-being during this critical period of global health. This needs to be considered in the discussion.

Another important point is that some participants received financial incentives (money or student credit) to participate, which may influence the motivation to participate in the study and its results. This needs to be discussed.

Something else that should be discussed as a limitation, given its implications for the observed results, is the differences in the instruments used for data collection between the studies, as well as the use of self-reported information such as weight and height, which may introduce some inaccuracies and important biases into the data.

Final decision

In light of the comments provided, I recommend that the requested changes and improvements be considered by the authors, and that after the corrections, the article be published.

**Do you want your identity to be public for this peer review?** For information about this choice, including consent withdrawal, please see our Privacy Policy

Reviewer #1: No

Reviewer #2: No

---

## [Author Response · Author response to Decision Letter 1]

15 Jul 2025

Journal Requirements:

1. When submitting your revision, we need you to address these additional requirements. Please ensure that your manuscript meets PLOS ONE's style requirements, including those for file naming. The PLOS ONE style templates can be found at https://journals.plos.org/plosone/s/file?id=wjVg/PLOSOne_formatting_sample_main_body.pdf and https://journals.plos.org/plosone/s/file?id=ba62/PLOSOne_formatting_sample_title_authors_affiliations.pdf

Author Response: Manuscript has been updated to fit style requirements as specified in the provided templates.

2. Thank you for uploading your study's underlying data set. Unfortunately, the repository you have noted in your Data Availability statement does not qualify as an acceptable data repository according to PLOS's standards. At this time, please upload the minimal data set necessary to replicate your study's findings to a stable, public repository (such as figshare or Dryad) and provide us with the relevant URLs, DOIs, or accession numbers that may be used to access these data. For a list of recommended repositories and additional information on PLOS standards for data deposition, please see https://journals.plos.org/plosone/s/recommended-repositories.

Author Response: Respectfully, we would like to keep the data in the Open Science Framework (OSF) because this Framework is used most extensively within psychology, and is actually listed as an acceptable option on the website you directed us to https://journals.plos.org/plosone/s/recommended-repositories. Moreover, PLOS policy states that “PLOS does not dictate repository selection for the data availability policy.”

3. We note that you have referenced (103). Conner TS, et al. Ultra-processed foods predict poorer mental health in young adults: a cross-sectional investigation in three countries. Unpublished manuscript. University of Otago, New Zealand; 2021.) which has currently not yet been accepted for publication. Please remove this from your References and amend this to state in the body of your manuscript: 103). Conner TS, et al. Ultra-processed foods predict poorer mental health in young adults: a cross-sectional investigation in three countries. Unpublished manuscript. University of Otago, New Zealand; 2021.) as detailed online in our guide for authors http://journals.plos.org/plosone/s/submission-guidelines#loc-reference-style.

Author Response: We have removed this from References and updated its removal in the main body of the manuscript.

4. We notice that your supplementary [tables] are included in the manuscript file. Please remove them and upload them with the file type 'Supporting Information'. Please ensure that each Supporting Information file has a legend listed in the manuscript after the references list.

Author Response: Supplementary tables have been removed from the manuscript file and are within the attached file “Supporting Information”. Legends for each of the tables are still present within the manuscript after the references list.

Reviewers' comments:

Reviewer's Responses to Questions

Comments to the Author

5. Review Comments to the Author

Reviewer #1: The explanation of the dual continuum models is not fully understood in the manner authors writing in the introduction. It is recommended that the authors re write the explanation in a more readable manner in the introduction section.

Author Response: The explanation of the dual continuum model in the introduction has been reworded/slightly expanded to improve readability.

The authors could relate that Baltic Sea diet as fish consumption in the introduction. Most of the readers are not familiar with this diet. Could they explain a little bit more about that diet before writing the results of the study.

Author Response: The explanation of what the Baltic Sea diet consists of has been expanded to mention the key food groups that comprises this diet.

The manuscript doesn’t contain the figures of the article.

Author Response: Journal formatting rules for submission meant the figures were not contained within the manuscript file, but attached as separate pngs/jpgs/etc that should have been available to reviewers. In the interest of readability, we have added in the figures into the revised manuscript file, in addition to submitting them as separate figure files. If this clashes with journal guidelines for submission, the editor is welcome to remove them.

The data analysis plan was really carefully done. The authors insisted on between subjects analysis and within subject analysis. But I am afraid the mixed model table is a bit hard to relate to that. I am referring to table 4 which is very condensed and full of information. Could the authors write a footnote or more explanation for their mixed model technique. I think the title of the graphs indicate the answer to some of my questions about the results section. I don’t have the graphs in my pdf sent from the journal.

Author Response: Thank you so much for noting that our data analysis plan was carefully done. Our team includes a biostatistician (RST), so we have focused on ensuring our modelling was done to a very high standard. Regarding Table 4, we have added a clarifying note describing our mixed model technique as well as brief explanations for the between- and within-person coefficients in the Table 4. Also, we are sorry you did not get the graphs. We have included them in the revised manuscript, in addition to submitted them as separate figure files.

The authors in discussion section reported that the physical activity was the least effective variable for well being based on their analysis and yet they say their results are aligned with others relating more sleep is linked with better mood. If the authors want to report this finding they should get the results section between good sleep quality and mood in the three data sets before jumping to this conclusion.

Author response: The author(s) are unsure which section of the discussion this is referring to; possibly lines 755-758 or 821-823. Both sections have been adjusted to use softer wording to avoid jumping to conclusions.

Reviewer #2: Decision Letter

Title: The title is somewhat sensationalist regarding the phrase "From surviving to thriving." I think it could perfectly well be just "How sleep, physical activity, and diet shape well-being in young adults." This is merely a suggestion, not an imposition, but I consider that "From surviving to thriving" feels disconnected from the article as a whole, as this statement is never revisited in the text, and the authors did not discuss in the discussion how well-being could be regained or improved based on the observed results. To include this element in the title, it would have been necessary to address how these lifestyle habits are basic human needs and how improving these factors can positively impact individuals' well-being, taking them from basic needs (surviving) to truly living well (thriving).

Author response: The author(s) have elected to retain the current title and have added some additional connection and context as suggested into the discussion (lines 724-726). Thank you.

Introduction: The introduction, although providing a good contextualization of the studied topic and good justifications for the study's objective, is unnecessarily lengthy. With so much information, the introduction feels long, almost giving the impression that much is already known about the subject, somewhat diminishing the need for another study on this topic. I suggest reducing the text a bit, removing some details about other studies, and leaving this type of comparison between previous studies more for the discussion.

Author response: The introduction has been shortened both by using more concise wording and moving some sections into the methods section.

The section of the introduction that talks about the objectives should conclude the introduction; there is no need to start detailing the datasets in the introduction, as this fits better in the methods section.

Author response: see above; the introduction now ends with the discussion of the objectives.

The paragraph starting at line 158 also seems somewhat unnecessary, as it tries to justify something already justified earlier about the importance of the "big three behaviors." I understand that this information strengthens the rationale for the choice of adjustment variables, but it does not need to be discussed so early in the article.

Author response: this paragraph has been moved to the Methods section where the health behaviours are first discussed.

Methods: The authors do not mention how participants were selected for the original studies. The lack of this information compromises the understanding of possible limitations and biases in the selection process, issues that may affect the representativeness and generalizability of the obtained results. I suggest that the authors briefly mention the type of method used to select participants for each dataset used, and whether participants were selected differently within the same dataset (convenience sample? random sampling?). Some parts of the methods suggest the use of convenience samples, but this needs to be clearer.

Author response: We now clarify that all three samples were convenience samples, but that Study 1 stratified recruitment by gender to achieve a more equal gender balance of men and women. Study 2 and 3 were strictly convenience samples (lines 161-163).

It is mentioned that in Study 1, participants who showed evidence of response bias were excluded. What would these response bias evidences used as criteria be? How was this evaluated and judged to decide who would be excluded for this reason?

Author response: While response bias can sometimes be spotted visually, to standardise the process we computed the intra-individual response variability index (IRV), a measure that captures atypical response patterns in a survey from the standard deviation of a long string of consecutive responses in a questionnaire. We calculated the IRV index for one of the longer measures in that survey (100-item personality measure) to flag extremely unusual response patterns such as straight-liners (consecutive identical responses like 3,3,3,3,3,3,3; i.e., IRV = 0) and individuals whose responses were extremely random (i.e., very high IRV). The IRV was calculated in R (R Core Team, 2013) using the ‘careless’ package (Yentes, 2021). Doing so identified cases to check by visual inspection of their data (very high or very low IRV values) and of these, four people were determined to have a response bias mainly due to low IRV values (e.g., selecting all 12 ethnicities as 1s, having no variability in measures where it is impossible to not have variability). We also identified four more people who had very extreme values on variables that were physically impossible, suggesting they were not reading or responding to the question correctly or just typing in values.

Dunn, A. M., Heggestad, E. D., Shanock, L. R., & Theilgard, N. (2018). Intra-individual response variability as an indicator of insufficient effort responding: Comparison to other indicators and relationships with individual differences. Journal of Business and Psychology, 33(1), 105–121. https://doi.org/10.1007/s10869-016-9479-0

R Core Team (2013). R: A language and environment for statistical computing. R foundation for statistical computing [Computer software]. https://www.R-project.org/.

Yetes, R. (2021). Package ‘careless’ [Computer software]. https://github.com/ryentes/careless/

The eligibility criteria of the original studies also need to be clarified, not just the reasons for exclusion from the analysis.

Author response: All three studies list the criteria participants needed to meet to be eligible for study participation (see Study 1-3 section of Participants and procedures). A more explicit description of eligibility criteria for Study 3 has been added (lines 217-220). There were no additional exclusion criteria.

In line 251, it is mentioned that 818 individuals aged between 17 and 25 years had usable data. This is confusing, as from the beginning (abstract and parts of the introduction), it is stated that participants were between 18 and 15 years old. If one of the datasets included eligible individuals aged 17, then the authors are indeed including individuals aged 17 to 25 in the total N. This needs to be clear from the beginning of the text, preferably from the abstract.

Author response: We have corrected this error. All descriptions of total N (including in the abstract) have been adjusted to reflect an age range of 17-25.

Results: In Table 1, I suggest presenting only the mean and standard deviation of age in each study. I see no reason to present it in categories, especially considering that some ages have very few or no participants, resulting in too much unused information in the table. I also suggest reviewing the rounding of decimal places, as some items do not add up to 100%.

Author response: Table 1 now only presents the mean/SD of age, and decimal places have been revised.

The way SES was assessed was not clear enough in the methods, making it somewhat difficult to understand what these values in Table 1 mean. How was this assessed? What do the mean and standard deviation in the table represent? I suggest explaining this better in the methods section or adding a footnote to facilitate interpretation in the table.

Author response: The item and scoring of the SES measure is now more detailed in the methods section.

In Table 2, I suggest removing the term "daily" from the title, as Study 1 does not use daily information.

Author response: removed.

As a general critique of all presented tables, the sample N used in the analyses should always appear, either in the table title, footnote, or within the table itself, as besides the differences in N between studies, it is necessary to be transparent about whether any data remained missing. The article is quite long, and it is difficult to keep searching for this information in other parts of the article to recall.

Author response: All presented tables now present the sample N relevant to the given table’s information.

Another important point: was presenting median and interquartile range considered for variables with skewed distributions? I ask because certainly some of the studied variables have a skewed distribution, and using the mean may not be ideal as it is influenced by extreme values. I suggest evaluating this point.

Author response: We now report the median and interquartile range for the Physical Activity variables (Table 2) in the descriptive statistics section. However, we have kept the analyses as is because the positive skew of this variable did not adversely impact the models due to the residuals from the models being normally distributed.

Discussion: Data from 2021 were used in Study 1. How might the results have been affected by the pandemic context, given that in 2021 the situation was still quite serious, and people had not yet fully resumed their routines and habits, especially regarding the three main behaviors studied? Study 3 also used data collected during a period when COVID-19 was still a major concern. Was this considered? It seems that the results obtained in Study 1, which observed higher consumption of ultra-processed foods, might be related to this data collection period, especially considering the observed correlation between this consumption and depressive symptoms. In many cases, individuals worsened their diet quality during the pandemic due to social isolation, anxiety, and distress during that period. It is possible that many people found some comfort in consuming these unhealthy foods, which might have brought some well-being during this critical period of global health. This needs to be considered in the discussion.

Author response: We now discuss COVID-19 disruptions in the Discussion. However, we do need to clarify that the ultra processed food (UPF) measure in Study 1 combined seven different food categories (summing weekly intake, and then dividing by 7 to approximate daily intake), whereas the daily UPF measure from Study 2 combined three food categories and Study 3 combined two food categories (both at the daily

---

## [Editor Report · Decision Letter 1]

21 Jul 2025

From surviving to thriving: How sleep, physical activity, and diet shape well-being in young adults

PONE-D-24-45325R1

Dear Dr. Conner,

We’re pleased to inform you that your manuscript has been judged scientifically suitable for publication and will be formally accepted for publication once it meets all outstanding technical requirements.

Kind regards,

Elma Izze Da Silva Magalhães

Academic Editor

PLOS ONE

---

## [Editor Report · Acceptance letter]

PONE-D-24-45325R1

PLOS ONE

Dear Dr. Conner,

I'm pleased to inform you that your manuscript has been deemed suitable for publication in PLOS ONE. Congratulations! Your manuscript is now being handed over to our production team.

Kind regards,

on behalf of

Dr. Elma Izze Da Silva Magalhães

Academic Editor

PLOS ONE